# MRN complex-dependent recruitment of ubiquitylated BLM helicase to DSBs negatively regulates DNA repair pathways

Vivek Tripathi[1], Himanshi Agarwal[1], Swati Priya[1], Harish Batra[1], Priyanka Modi[1], Monica Pandey[2], Dhurjhoti Saha[3], Sathees C. Raghavan[2] & Sagar Sengupta[1]

Mutations in BLM in Bloom Syndrome patients predispose them to multiple types of cancers. Here we report that BLM is recruited in a biphasic manner to annotated DSBs. BLM recruitment is dependent on the presence of NBS1, MRE11 and ATM. While ATM activity is essential for BLM recruitment in early phase, it is dispensable in late phase when MRE11 exonuclease activity and RNF8-mediated ubiquitylation of BLM are the key determinants. Interaction between polyubiquitylated BLM and NBS1 is essential for the helicase to be retained at the DSBs. The helicase activity of BLM is required for the recruitment of HR and c-NHEJ factors onto the chromatin in S- and G1-phase, respectively. During the repair phase, BLM inhibits HR in S-phase and c-NHEJ in G1-phase. Consequently, inhibition of helicase activity of BLM enhances the rate of DNA alterations. Thus BLM utilizes its pro- and anti-repair functions to maintain genome stability.

[1] National Institute of Immunology, Aruna Asaf Ali Marg, New Delhi 110067, India. [2] Department of Biochemistry, Indian Institute of Science, Bangalore 560012, India. [3] Institute of Genomics and Integrative Biology, CSIR, Mathura Road, New Delhi 110025, India. Correspondence and requests for materials should be addressed to S.S. (email: sagar@nii.ac.in)

DNA damage response (DDR) is an integrated and choreographed response mounted by the cells to eliminate multiple forms of DNA damage that occur in the form of double-strand breaks (DSBs) and stalled replication forks, thereby allowing the faithful replication of the genome. In contrast to stalled replication forks that are recognized by the ATR/Chk1 axis, DSBs are known to be recognized by MRE11–RAD50–NBS1 (MRN) complex, which recruits ATM and allows its optimal activation. ATM, in turn phosphorylates NBS1 leading to the stabilization of MRN complex and subsequently the sequential accumulation of other DDR proteins, thereby allowing the progression of the DDR (reviewed in ref. [1]).

Mutations in tumor suppressor BLM helicase cause Bloom Syndrome (BS). BS patients are characterized by predisposition to multiple types of cancers[2]. Experiments with replication stress have provided evidence that BLM plays an important role in the recognition and subsequent resolution of this type of DNA damage[3]. BLM is an early responder to the formation of stalled replication forks[4] and is recruited to them after being ubiquitylated at three lysine residues by the two chromatin bound E3 ligases, RNF8 and RNF168[5]. The hierarchical position of BLM during replication stress is well deciphered. ATR- and ATM-dependent recruitment of BLM at these sites ensures subsequent ATM activation and 53BP1 focus formation[6]. BLM is essential for the recruitment of MRN complex and BRCA1 to the stalled replication forks[7,8]. However less is known about how BLM is recruited to another more prevalent, potent, and physiological class of lesions, the DSBs. It is known that BLM is an early responder to DSBs[9]. Further chromatin loading of BLM to the IR-induced DSBs depends on the presence of Rap1-interacting factor 1 (RIF1) and RNA helicase DEAD Box 1 (DDX1)[10,11]. ATM-mediated phosphorylation of CtIP also promotes the recruitment of BLM to laser-induced DSBs[12].

Once recruited to the site of damage, BLM is involved in its repair. The multiple mechanisms by which BLM functions during homologous recombination (HR), particularly the requirement of its post-translation modifications, have been well documented[13]. For example, BLM is instrumental in the dissolution of both RAD51 nucleoprotein filaments[14–16] and double Holliday junctions[17]. BLM has also been implicated in the negative regulation of error-prone microhomology-mediated end joining (MMEJ). Hence, in cells lacking BLM, the rate of MMEJ-mediated genomic rearrangements was enhanced, indicating BLM prevents this inaccurate pathway of DSB repair[18–20].

In recent past, site-specific cleavage in the human genome has been achieved by fusing the restriction enzyme AsiSI to a modified estrogen receptor (ER) hormone-binding domain, which only binds to 4-hydroxy tamoxifen (4-OHT), an active metabolite of tamoxifen. This has led to the generation of the U2OS–AsiSI–ER system, which has been used previously to obtain the high-resolution profiling of γH2AX around DSBs in the human genome. It was inferred that γH2AX distribution was dependent on gene transcription[21].

Using this U2OS–AsiSI–ER system we have now deciphered the key determinants essential for the recruitment of BLM to the DSBs. We provide evidence that the recruitment of BLM occurs in a biphasic manner extending from 80 bp from the DSBs to 9 kb from the DSBs. While BLM recruitment does not require on its own helicase activity, it is dependent on the kinase activity of ATM and the exonuclease activity of MRE11. Another key requirement is the interaction of RNF8 ubiquitylated BLM with a second MRN complex member, Nbs1. The retention of BLM at the DSBs coincides with its co-recruitment with HR factors in S-phase and c-NHEJ components in G1-phase. The helicase activity of BLM is critical for the phase-specific recruitment of the repair factors to the DSBs. In this phase, BLM recruitment is independent of ATM kinase activity but still depends on the MRE11 exonuclease activity. Post recruitment during the repair phase, BLM not only inhibits HR in S-phase, but also c-NHEJ in G1-phase. Further c-NHEJ is inhibited in the S-phase as a backup mechanism. Thus BLM both enhances and inhibits multiple repair mechanism in a cell cycle phase-dependent manner and thereby maintains genome integrity.

## Results

**Recruitment of BLM to DSBs is biphasic in nature.** In an effort to understand the mechanism of BLM recruitment at the DSBs, we treated U2OS–AsiSI–ER cells with 4-OHT for different time intervals upto 4 h (details of all treatments in the recruitment phase in Supplementary Figure 1A). The extent of DSBs increased with time as indicated both by the levels of the DDR proteins (γH2AX, NBS1, pSer1981 ATM) and their foci formation (Supplementary Figure 2A, C). Concomitant with other DDR factors, the levels of endogenous BLM also increased after DSB generation as early as 30 min after 4-OHT treatment which became more pronounced after 4 h (Supplementary Figure 2A). BLM colocalized with NBS1, γH2AX, and pSer1981 ATM foci (Fig. 1a, Supplementary Figure 2B), further indicating that BLM is an integral component of the DDR. The colocalization between BLM and γH2AX foci was observed as early as 30 min post-damage induction and increased with time (Supplementary Figure 2C).

To elucidate whether BLM is recruited to annotated DSBs formed after 4-OHT treatment, we carried out chromatin immunoprecipitation (ChIP) with anti-BLM antibody in U2OS–AsiSI–ER cells treated with 4-OHT for 4 h. BLM recruitment to both proximal and distal regions to the DSBs on multiple chromosomes (annotated positions in Supplementary Table 1) was determined by ChIP-qPCR by an anti-BLM antibody that did not detect any BLM recruitment when BLM levels are depleted by the cognate BLM siRNA (Supplementary Figure 2D). DSB induction caused BLM recruitment to all the tested seven DSBs spread across different chromosomes (Fig. 1b, Supplementary Figure 2D, E).

To determine whether BLM recruitment was dependent on the compactness of the chromatin, two histone marks, H3K36me3 (associated with euchromatin) and H3K9me3 (associated with heterochromatin), were analyzed utilizing the ChIP-seq data for U2OS cells from the ENCODE database. Analysis of the chromatin landscape on either side of multiple AsiSi sites (by superimposing the primers used for ChIP-qPCR with the ChIP-seq data) revealed that BLM recruitment occurred irrespective of the compactness of the surrounding chromatin (Supplementary Figure 3).

To determine whether the recruitment of BLM was independent of its own helicase function, BLM ChIP was carried out without or in presence of ML216, a specific inhibitor for BLM helicase activity[22] as validated by the SCE assay (Supplementary Figure 2F, G). Presence of ML216 did not prevent BLM from being recruited to the DSBs (Fig. 1c), indicating other transacting factors regulate its recruitment.

To determine the spacial and temporal recruitment of BLM, U2OS–AsiSI–ER cells were exposed to 4-OHT for different time intervals (0.5–4 h) and the extent of recruitment of BLM was determined at different distances (80 bp to 9 kb) from the DSBs for two randomly picked AsiSI sites (annotated positions in Supplementary Table 2). BLM was recruited to the DSBs in a biphasic manner. The first wave of BLM recruitment to the AsiSI-mediated DSBs occurred within 0.5 h, followed by a pronounced decrease in the extent of recruitment upto 2 h. However, 4 h post-DSB induction again led to a robust BLM recruitment onto the chromatin. Interestingly BLM recruitment occurred to roughly

similar extent irrespective of the distance from the DSBs (Fig. 1d, Supplementary Figure 4A).

The biphasic recruitment of BLM indicates that BLM probably has roles in both the early (i.e., the damage sensing step) and late (i.e., the damage repair phase) phase post-DSB generation. To characterize the early recruitment in more detail, we carried out parallel ChIPs with anti-BLM, anti-NBS1, anti-XRCC4, and anti-RAD51 antibodies after exposing the cells to 0.5 h of 4-OHT treatment. It was observed that after 0.5 h of 4-OHT treatment both BLM and NBS1 are co-recruited to the DSBs. The extent of BLM and NBS1 recruitment was higher than RAD51 and XRCC4. This indicated that at this early time point, the DSBs were

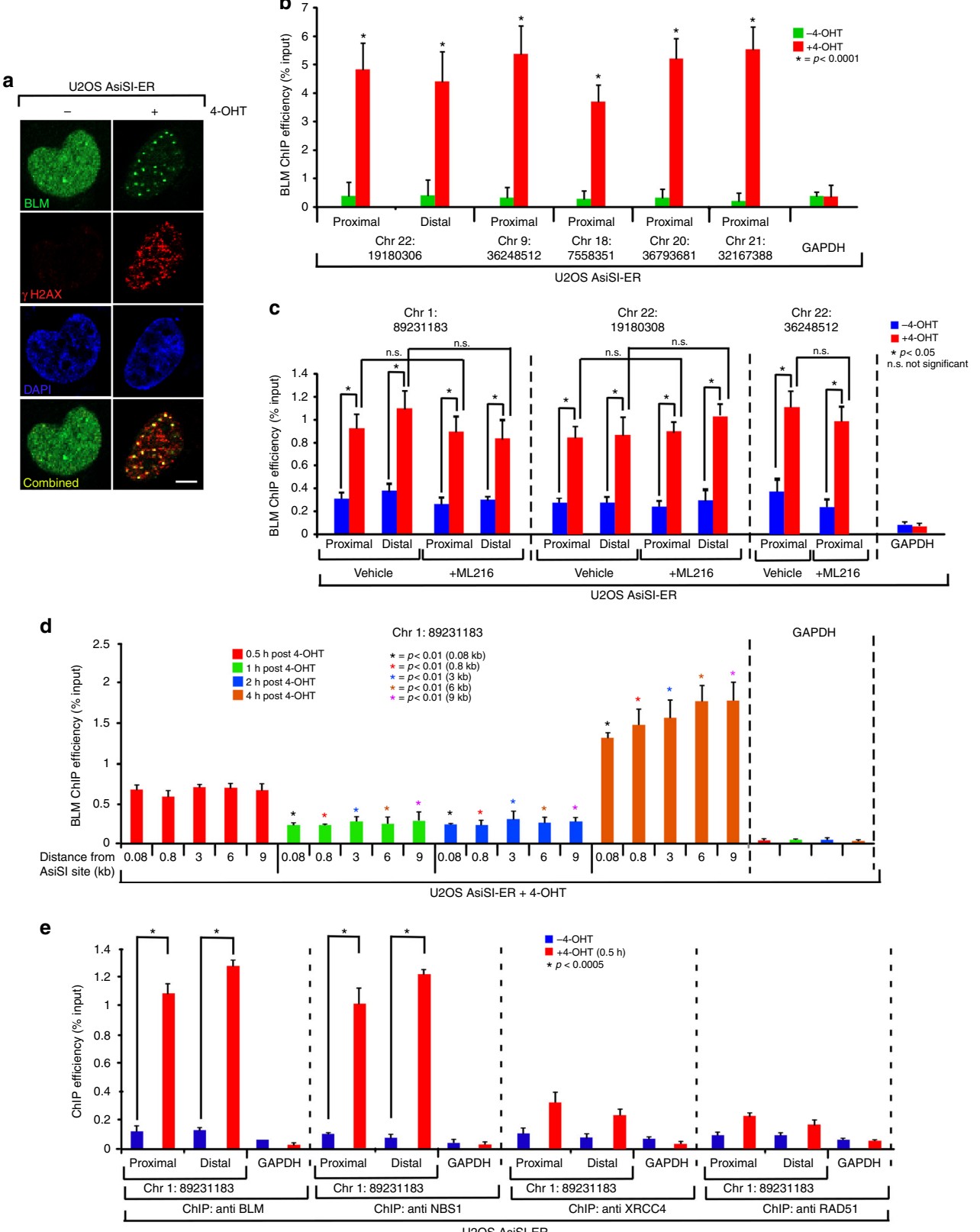

recognized by the DNA damage sensor proteins like NBS1. At this time window, DNA repair processes have not been fully initiated (Fig. 1e, Supplementary Figure 4B).

**MRN complex and ATM control BLM recruitment to the DSBs.** Since BLM recruitment did not depend on its own helicase activity (Fig. 1c), we wanted to determine the trans-regulatory factors essential for its recruitment to the DSBs. Indeed immunoprecipitations carried out with anti-BLM and anti γH2AX antibodies revealed that co-immunoprecipitation of BLM with pSer1981 ATM, all members of the MRN complex (MRE11, RAD50, and NBS1) and γH2AX (Fig. 2a, Supplementary Figure 5B) increased after 4-OHT treatment in U2OS–AsiSI–ER cells. siRNA mediated depletion of NBS1, MRE11, or ATM (Supplementary Figure 5C, D) abrogated both early (i.e., at 0.5 h) and late (i.e., at 4 h) phase BLM recruitment to both the proximal or distal regions of multiple AsiSI sites after 4-OHT treatment (Fig. 2b, c, Supplementary Figure 6, 7).

Further, to determine whether BLM recruitment at the DSBs depended on the enzymatic activities of MRE11 or ATM, BLM ChIP was carried out after DSB induction, in the absence or presence of either MRE11 exonuclease activity inhibitor (Mirin) or ATM kinase inhibitor (KU 55933). Treatment with either Mirin or KU 55933 prevented the phosphorylation of multiple known ATM substrates (Supplementary Figure 8A, B). In the presence of Mirin, both early (i.e., at 0.5 h) and late (i.e., at 4 h) phase BLM recruitment to all the tested DSBs was decreased by ~50% (Fig. 2d, Supplementary Figure 8C), suggesting that the single-stranded DNA generated by MRE11 exonuclease activity is a key determinant. Interestingly, while the recruitment of BLM in the early phase was dependent on ATM activity, BLM was present on the chromatin even in absence of ATM activity after 4 h of 4-OHT treatment (Fig. 2e, Supplementary Figure 8D). These results indicated that in addition to the physical ablation of the proteins, the loss of exonuclease activity of MRE11 and kinase activity of ATM regulated early BLM recruitment to the DSBs. However, in the second wave of BLM recruitment the requirement of ATM activity is not there indicating the contribution of other factors at this step.

**Interaction between BLM and MRN complex in recruitment phase.** Based on the present results, we wanted to determine whether MRN-dependent recruitment of BLM after DSB generation also involved its own K63-linked ubiquitylation, as previously shown for replication stress[5]. A low-affinity direct physical interaction between BLM and NBS1 was detected in vitro (Supplementary Figure 9). However, the interaction between RNF8-dependent polyubiquitylated BLM (BLM WT) with immunoprecipitated NBS1 was much stronger compared to non-ubiquitylated BLM (BLM 3K) (Fig. 3a, b, bottom panels). This indicated that BLM polyubiquitylation is the key determinant for its interaction with NBS1. Further in cells lacking BLM, exogenous wild-type BLM but not BLM 3K interacted with endogenous NBS1 (Fig. 3c). Hence lack of BLM polyubiquitylation in vivo (due to depletion of RNF8 by siRNA, Fig. 3d, left), abolished BLM/NBS1 interaction (Fig. 3d, right) in U2OS–AsiSI–ER cells after 4 h of DSB induction. Interaction of polyubiquitylated BLM occurs only with NBS1 and not MRE11 (Fig. 3e). Thus the effect of MRE11 (seen earlier with siMRE11 and Mirin) on BLM recruitment (Fig. 2b, d, Supplementary Figure 6, 8C) is not due to a direct physical interaction. Instead, polyubiquitylated BLM and NBS1 interaction is the key step which brings MRE11 in contact with BLM, allowing BLM to be retained at the DSBs by utilizing MRE11's exonuclease activity.

The above results indicate that ubiquitylation of BLM by RNF8 and the exonuclease activity of MRE11 are possibly two key enzymatic determinants regulating the second wave of BLM recruitment. To test this hypothesis, BLM recruitment to the DSBs was determined in U2OS AsiSI–ER cells after 4 h of 4-OHT treatment by ChIP (Fig. 3f) or foci formation (Fig. 3g, h). The experiment was done in presence of siRNF8 alone, or in cells treated with Mirin or in absence of both RNF8 and the exonuclease activity of MRE11. Lack of either RNF8 (Supplementary Figure 5E) or MRE11 exonuclease activity due to Mirin treatment partially decreased the recruitment of BLM to both the proximal and distal regions of all tested DSBs and partially affected the consequent appearance of the BLM foci. However, depletion of both ubiquitylation and MRE11 exonuclease activities completely abolished BLM recruitment to the DSBs (Fig. 3f–h), suggesting that RNF8-mediated ubiquitylation along with MRE11 exonuclease activity are key events that allows BLM to be recruited to the chromatin.

**BLM recruits HR and c-NHEJ factors to chromatin.** In an effort to determine the effect of BLM recruitment to the DSBs, we wanted to dissect how BLM affects the two main repair pathways (HR and c-NHEJ) in S- and G1-phase of the cell cycle. ChIPs using BLM, RAD51, and XRCC4 antibodies were carried out in U2OS–AsiSI–ER cells synchronized either in S- or G1-phase of the cell cycle by double thymidine block and subsequently released (Supplementary Figure 10C). The cells in the S- or G1-phase were either left untreated (−4-OHT) or treated with 4-OHT for 4 h. During the S-phase, BLM (Fig. 4a, Supplementary Figure S10B) and RAD51 (Fig. 4b, Supplementary Figure 10C)

**Fig. 1** BLM is recruited to AsiSI-induced DSBs. **a** BLM and γH2AX colocalize after DSB induction. U2OS–AsiSI–ER cells were grown asynchronously (−4-OHT) or in the presence of +4-OHT for 4 h. Cells were fixed and stained with anti-BLM and anti-γH2AX antibodies. Nucleus was stained by DAPI. Bar, 5 μM. **b** BLM is recruited to sequence-specific DSBs. U2OS–AsiSI–ER cells were grown asynchronously (−4-OHT) or in the presence of +4-OHT for 4 h. ChIP was carried out using anti-BLM antibody. Recruitment of BLM to the indicated AsiSI-induced DSBs or GAPDH loci were determined by ChIP-qPCR carried out using primers, which were either proximal or distal to the AsiSI sites. **c** Recruitment of BLM to the DSBs is helicase independent. Same as **b**, except BLM recruitment to the indicated AsiSI-induced DSBs was determined in U2OS–AsiSI–ER cells grown either in the absence or presence of ML216. 4-OHT treatment was for 4 h. **d** Biphasic BLM recruitment occurs at both proximal and distal positions with respect to the AsiSI site. Same as **b**, except BLM recruitment to the indicated AsiSI-induced DSB was determined after 4-OHT treatment for 0.5, 1, 2, and 4 h. ChIP was carried out using anti-BLM antibody. Recruitment of BLM to the indicated AsiSI-induced DSBs or GAPDH loci were determined by ChIP-qPCR. Distance from AsiSi site at which recruitment was measured was approximately 0.08, 0.8, 3, 6, and 9 kb. **e** BLM is co-recruited with NBS1 but not with DNA repair proteins to the DSBs in early phase of recruitment. Same as **b** except U2OS–AsiSI–ER cells were treated with 4-OHT for 0.5 h. Parallel ChIPs were carried out with anti-BLM, anti-NBS1, anti-XRCC4, and anti-RAD51 antibodies. Recruitment of BLM, NBS1, XRCC4, and RAD51 to the indicated DSBs or to the GAPDH loci was determined by carrying out ChIP-qPCR analysis. For all ChIP-qPCR analysis, the depicted values (mean ± standard deviation) were obtained from four independent experiments. Data were analyzed by unpaired two-tailed Student's *t*-test

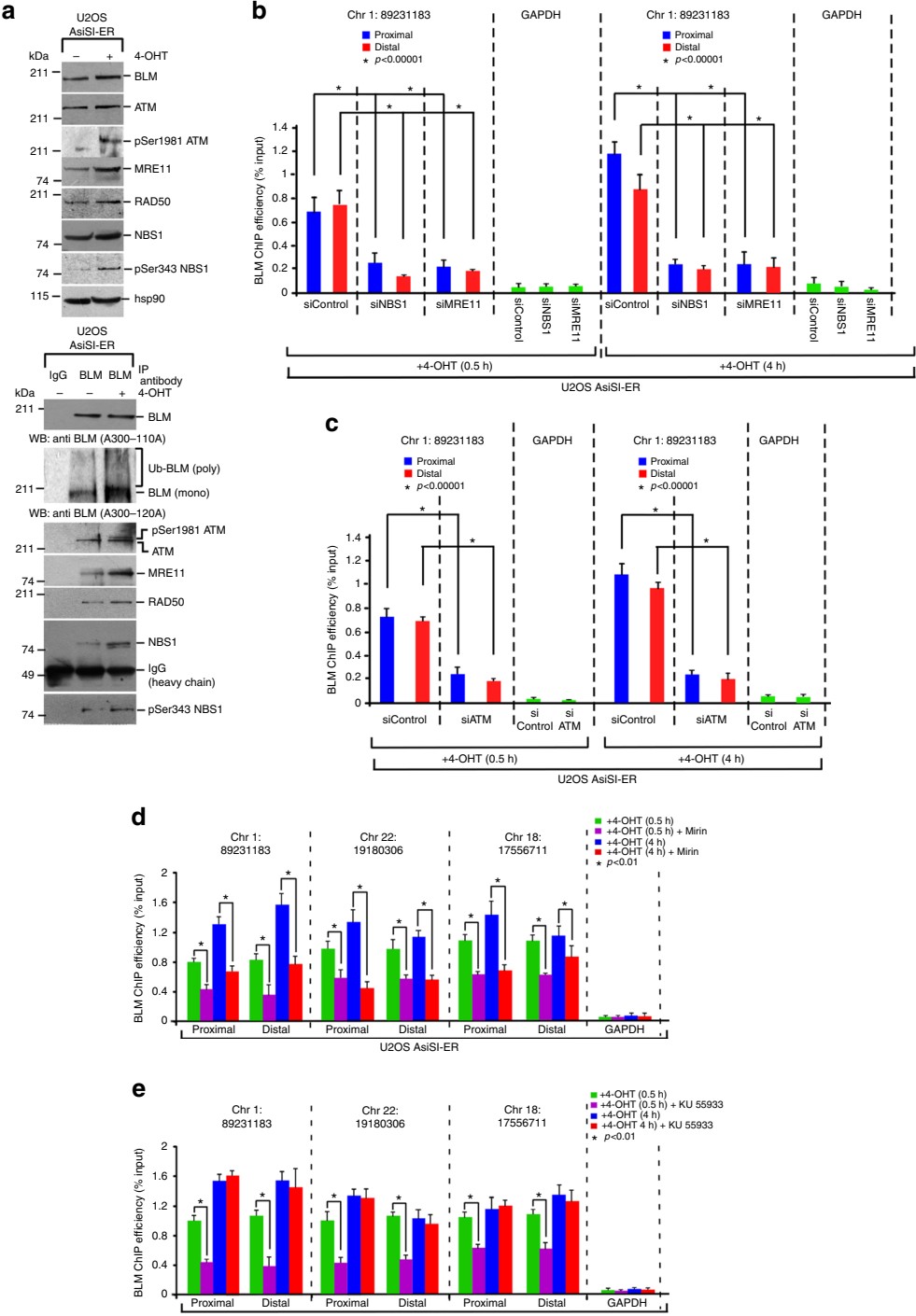

**Fig. 2** Recruitment of BLM at the DSBs. **a** BLM co-immuoprecipitates with MRN complex and ATM after DSB generation. (Top) Extracts were made from U2OS–AsiSI-ER cells (±4-OHT, 4 h). Lysates were probed with antibodies against BLM (only A300-110A), ATM, pSer1981 ATM, MRE11, RAD50, NBS1, pSer343 NBS1, hsp90. (Bottom) Immunoprecipitations (±4-OHT, 4 h) were carried out with anti-BLM antibody or the corresponding IgG and the immunoprecipitates were probed with the above-mentioned antibodies. **b**, **c** Recruitment of BLM to the DSBs is dependent on the presence of NBS1, MRE11, and ATM in both 0.5 h and 4 h post induction of DSBs. ChIP was carried out using anti-BLM antibody on chromatin obtained from U2OS–AsiSI–ER cells grown in the presence of 4-OHT for either 0.5 or 4 h after transfection with either **b** Control siRNA or NBS1 siRNA or MRE11 siRNA, or **c** control siRNA or ATM siRNA. BLM recruitment to the indicated AsiSI-induced DSB or GAPDH loci were determined by ChIP-qPCR after either 0.5 or 4 h of 4-OHT treatment. **d** Recruitment of BLM to DSBs partially depends on MRE11 exonuclease activity. ChIP was carried out using anti-BLM antibody on chromatin obtained from U2OS–AsiSI-ER cells grown in presence of 4-OHT for either 0.5 or 4 h. In both the time points, the cells were grown either in +4-OHT or in +4-OHT + Mirin conditions. BLM recruitment to the indicated AsiSI-induced DSBs or GAPDH loci were determined by ChIP-qPCR. **e** Recruitment of BLM to DSBs depends on ATM kinase activity only in the early phase. ChIP was carried out using anti-BLM antibody on chromatin obtained from U2OS–AsiSI-ER cells grown in the presence of 4-OHT for either 0.5 or 4 h. In both the two time points, the cells were grown in +4-OHT or +4-OHT + KU 55933 conditions. BLM recruitment to the indicated AsiSI-induced DSBs or GAPDH loci were determined by ChIP-qPCR. For all ChIP-qPCR analyses, the depicted values (mean ± standard deviation) were obtained from four independent experiments. Data were analyzed by unpaired two-tailed Student's $t$-test

were recruited to both the proximal and distal regions of all the tested DSBs after 4-OHT treatment. In this phase, a low but detectable level of XRCC4 recruitment to the DSBs was also observed (Fig. 4c, Supplementary Figure 10D). Interestingly, BLM was also recruited during G1-phase to the proximal and distal regions of all the DSBs (Fig. 4d, Supplementary Figure 10E) along with XRCC4 (Fig. 4f, Supplementary Figure 10G), but not RAD51 (Fig. 4e, Supplementary Figure 10F). Further, both reciprocal immunoprecipitations and immunofluorescence experiments (Supplementary Figure 11B-E) indicated that BLM interacted with RAD51 during S-phase, while it complexed with XRCC4 in the G1-phase. A minor interaction of BLM with XRCC4 was reproducibly observed in S-phase in both these assays.

We next wanted to determine whether the helicase activity of BLM was essential for the recruitment and retention of the HR

and NHEJ factors to the chromatin after induction the DSBs in either G1- or S-phase of the cell cycle. ChIP-qPCR indicated that the absence of BLM helicase activity due to ML216 treatment prevented the optimal recruitment of RAD51 and XRCC4 in S- and G1-phase, respectively (Fig. 4g, h, Supplementary Figure 12). Fractionation studies further indicated that inhibition of BLM helicase activity decreased association of multiple HR factors in S-phase (Fig. 4i) and c-NHEJ factors in G1-phase (Fig. 4j) with the chromatin.

**BLM negatively regulates HR and c-NHEJ.** Having established the mechanism of BLM recruitment to annotated DSBs (Figs. 1–4), we wanted to determine whether BLM could also specifically regulate the two repair processes at these DSBs in a cell cycle-specific manner. To specifically study the role of BLM helicase

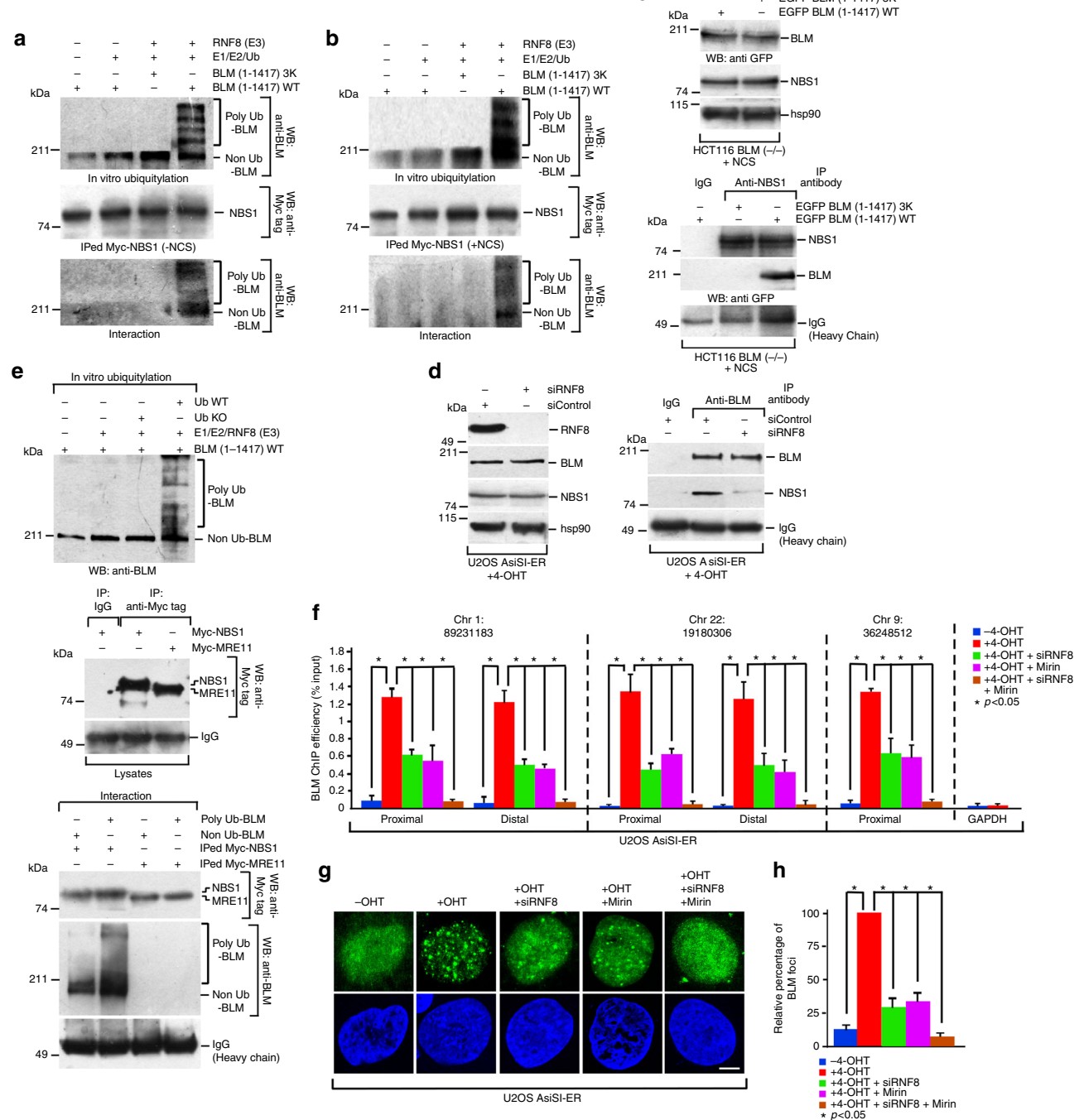

activity during the repair step, DSBs were induced by 4-OHT treatment for 4 h in either S- or G1-phase. Subsequently 4-OHT was washed off and the cell growth was continued for further 1 h (to allow the repair to happen) in absence of 4-OHT but in presence of ML216 or B02 (RAD51 inhibitor that disrupts the binding of RAD51 to DNA)[23] or SCR7 (Ligase IV inhibitor that blocks Ligase IV-mediated end joining by interfering with the DNA binding of Ligase IV)[24] (details of all treatments in the repair phase in Supplementary Figure 1B). During S-phase lack of active RAD51 due to the B02 treatment, decreased the rate of HR to the basal level. Inhibition of BLM activity by ML216 treatment caused an increase in the rate of HR. This negative effect of BLM on HR was not in G1-phase due to the lack of sister chromatids (Fig. 5a). In G1-phase, the major pathway of DNA repair is via NHEJ. Using an intrachromosomal substrate, total NHEJ increased in cells lacking active BLM due to ML216 treatment (Fig. 5b, Supplementary Figure 13A-C). Moreover, ML216 treatment increased the rate of joined product formation, indicating BLM negatively regulated c-NHEJ in G1 phase. As expected, treatment of cells with SCR7 abolished c-NHEJ, while the treatment with B02 forced the cells in S-phase to forgo HR and instead adapt c-NHEJ as the repair pathway. Interestingly, treatment of ML216 also increased c-NHEJ in S-phase. This indicated that though HR is the pathway preferentially inhibited by BLM, it (i.e., BLM) also exerts its negative regulatory effect on c-NHEJ during S-phase (Fig. 5c, d). The presence of both ML216 and SCR7 completely abrogated c-NHEJ in G1-phase, further validating that BLM negatively regulates c-NHEJ (Supplementary Figure 13D, E).

To conclusively demonstrate BLM's role in c-NHEJ, genomic DNA was isolated from G1-phase cells treated with either 4-OHT alone or a combination of 4-OHT and ML216. For either condition, 140 individual clones (35 each from four different AsiSI sites) were sequenced and the extent of different types of nucleotide changes quantitated (Fig. 5e, Supplementary Table 3). Absence of BLM helicase activity statistically increased the rates of sequence alterations (involving deletions, insertions, and deletions with insertions) at or near the AsiSI junctions, thereby providing further evidence that the absence of the helicase activity of BLM negatively affects c-NHEJ in G1-phase.

## Discussion

In this report, we delineate the mechanism of BLM recruitment to annotated DSBs spread throughout the human genome and its subsequent effect on DNA repair at these specific sites (Fig. 6). We provide evidence that BLM gets recruited to the DSBs in a biphasic manner. The distance of BLM recruitment ranges from ~80 bp to 9 kb with respect to the DSBs (Fig. 1d, Supplementary Figure 4A). At the DSBs, BLM is co-recruited with multiple components of the DDR pathway via a temporal and cell cycle phase-specific mechanism (Figs. 1e, 4a–f, Supplementary Figure 4B, 10B-G). BLM accumulates with almost equal efficiency on multiple DSBs located on different chromosomes (Fig. 1b, Supplementary Figure 2D, E). It was also observed that BLM recruitment does not depend on the nature of the surrounding chromatin architecture, as recruitment occurs irrespective of whether repressing or activating histone marks surround the AsiSI sites (Supplementary Figure 3). It will be interesting to compare the above mode of BLM recruitment to the recruitment profile of other DDR factors using the same U2OS-AsiSI–ER system. It has been reported that while γH2AX is depleted at the DSBs, it instead spreads to megabase distances away from the break site[21]. In contrast proteins do get recruited only to the immediate vicinity of the DSBs, such as pSer11981ATM[25] and also proteins involved in HR and NHEJ pathway namely RAD51, XRCC4[26]. Hence BLM recruitment seems to be unique as (a) it is present both in the immediate vicinity and far from the DSBs (Fig. 1b, d, Supplementary Figure 2D, E, 4A), (b) has a biphasic recruitment profile (Fig. 1d, Supplementary Figure 4A), and (c) accumulates irrespective of the chromatin architecture (Supplementary Figure 3).

Based on the above recruitment characteristics, BLM's involvement in DDR seems to occur at two distinct phases. In the early phase, BLM is co-recruited with DNA damage sensor proteins like NBS1, γH2AX, pSer1981 ATM within 0.5 h after induction of DNA damage (Fig. 1a, e, Supplementary Figure 2C, 4B). BLM recruitment in this phase occur irrespective of the chromatin architecture (Supplementary Figure 3) and possibly helps DDR by enhancing the functions of histone chaperones[27] and chromatin remodelers[28]. In the second phase of recruitment (4 h post DSB induction), BLM is present at the DSBs along with DNA repair proteins like XRCC4 and RAD51 (Fig. 4a–f, Supplementary Figure 10B-G, 11). In fact BLM is essential for the recruitment of multiple HR and NHEJ proteins to the chromatin (Fig. 4g–j, Supplementary Figure 12), thereby acting as a pro-repair protein. Once recruited, during the repair phase BLM negatively regulates both HR and c-NHEJ (Fig. 5, Supplementary Figure 13, Supplementary Table 3), thereby acting as an anti-repair protein. This dual functions of BLM, as both pro- and anti-repair factor, had been hypothesized earlier based on in vitro experiments[14]. We now provide evidence that these two activities of BLM indeed occur in vivo at two distinct stages in response to DNA damage.

**Fig. 3** Ubiquitylated BLM at the DSBs interacts with NBS1. **a, b** RNF8-dependent polyubiquitylation of BLM is required for its interaction with NBS1. (Top) **a, b** In vitro ubiquitylated BLM WT or BLM 3K was probed with anti-BLM antibody. (Middle) **a, b** Exogenous NBS1, immunoprecipitated from HEK293T cells (±NCS), was probed with anti-Myc tag antibody. (Bottom) **a, b** Interaction carried out with either polyubiquitylated or non-ubiquitylated BLM and bound Myc-tagged NBS1 was detected with anti-BLM antibody. **c** Ubiquitylation is essential for BLM to interact with NBS1 in vivo. HCT116 BLM (−/−) cells (+NCS) were transfected with either EGFP-C1 BLM (WT) or (3K). (Top) Lysates were probed with antibodies against GFP, NBS1, hsp90. (Bottom) Lysates were immunoprecipitated with antibody against NBS1 and probed with anti-NBS1 and anti-GFP antibodies. **d** Lack of RNF8 prevents the interaction between BLM and NBS1. (Left) U2OS–AsiSi-ER cells were transfected with siControl or siRNF8. Extracts were probed with anti-RNF8, anti-BLM, anti-NBS1, anti-hsp90 antibodies. (Right) Immunoprecipitations, carried out with either anti-BLM antibody or IgG, were probed with anti-BLM, anti-NBS1 antibodies. **e** Polyubiquitylated BLM is complexed with NBS1 but not MRE11. (Top) RNF8-dependent in vitro ubiquitylation of BLM WT, in the presence of Ub WT or Ub with all lysines mutated (Ub KO), was detected using anti-BLM antibody. (Middle) Exogenous Myc-tagged NBS1 and MRE11 were immunoprecipitated and detected with anti-Myc tag antibody. (Bottom) Interactions were carried out with either ubiquitylated or non-ubiquitylated BLM with immunoprecipitated Myc-tagged NBS1 or MRE11 and detected with anti-BLM antibody. **f** Recruitment of BLM to DSBs is abrogated in the presence of Mirin and siRNF8. ChIP was carried out using anti-BLM antibody on chromatin obtained from U2OS AsiSI-ER cells (4 h, 4-OHT). BLM recruitment to the indicated AsiSI-induced DSBs or GAPDH loci were determined by ChIP-qPCR. **g, h** Formation of BLM foci are abrogated in presence of Mirin and siRNF8. Experiments were done in same conditions as in **f**. **g** Immunofluorescence was carried out with anti-BLM antibody. Nucleus was stained with DAPI. Bar, 5 μM. **h** Quantitation of BLM foci formed under different conditions. Data were analyzed by unpaired two-tailed Student's t-test. Values mean ± standard deviation

The molecular basis that causes this switch in BLM function is yet unknown.

The recruitment of BLM to the DSBs depends on multiple *cis*- and *trans*-factors. *Cis*-factor like the helicase activity of BLM is not essential for its recruitment (Fig. 1c). *Trans*-factors like the physical presence of the MRN complex (specifically NBS1 and MRE11) and ATM protein are the essential prerequisites for the recruitment process (Fig. 2b, c, Supplementary Figure 6, 7). The

exonuclease function of MRE11 is essential for the complete recruitment of BLM in both the early and late phase after DSB induction (Fig. 2d, Supplementary Figure 8C). It has been earlier reported that BLM and MRE11 are both part of two end resection complexes that are involved in DNA break repair[29]. Thus the dependence of BLM recruitment to the chromatin after DNA damage on MRE11 expands the role of this exonuclease and indicates that BLM and MRN complexes have multiple functional

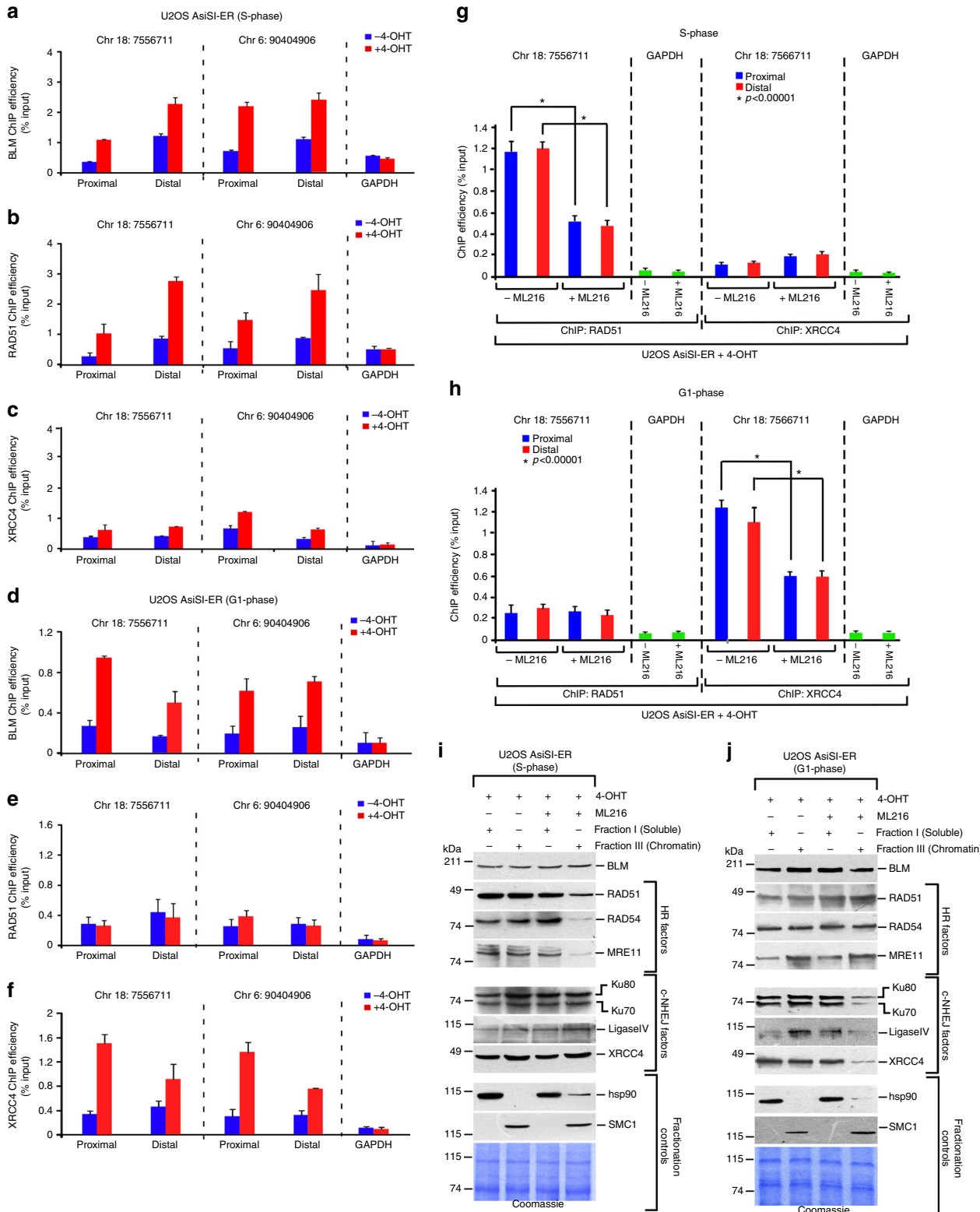

interactions during DNA damage response. In contrast, the recruitment of BLM to the chromatin depends on ATM only in the early phase, i.e., 0.5 h after DSB induction (Fig. 2e), as also observed earlier using laser-induced DSB generating system[12]. BLM dependence on ATM activity maybe direct—as BLM has already been shown to be an ATM substrate[30]. However, it is equally plausible that this is an indirect effect as ATM is known to phosphorylate and thereby regulate the recruitment of both NBS1 and MRE11 onto the chromatin[31]. In the late phase (i.e., 4 h after 4-OHT treatment), the co-recruitment of BLM with DNA repair factors XRCC4 and RAD51 (Fig. 4a–f, Supplementary Figure 10B-G) is independent of ATM kinase activity (Fig. 2e, Supplementary Figure 8D). In this phase, RNF8-mediated BLM ubiquitylation that allows BLM to interact with NBS1, along with the exonuclease activity of MRE11 are the two key essentials for BLM recruitment (Fig. 3f–h).

It is to be noted that while depletion of ATM leads to abrogation of BLM recruitment after both 30 min and 4 h of 4-OHT treatment, inhibition of ATM activity has its effect only at the earlier time point (Fig. 2c, e, Supplementary Figure 7, 8D). This indicates that at the later time periods of DSB induction, BLM that is already at the DSBs do not directly need the kinase activity of ATM. Instead the kinase activity of ATM in this time period is probably needed by its other substrates of the DDR pathway (like MRN complex)[31]. However, the physical presence of ATM is necessary, as without it these other substrates will not get phosphorylated and the DDR pathway cease to be effective. It is interesting to note that though BLM recruitment does not depend on ATM activity after 4 h of 4-OHT treatment, its phosphorylation by ATM at Thr99 might play a role in the repair phase as deciphered from the phenotypes associated with this BLM mutation[30]. Together these results indicate that the interaction of BLM with members of the DDR pathway occurs in both spatial and temporal context.

Interaction between BLM and NBS1 is a key component of BLM recruitment—both being part of the same DDR complex after DNA damage induction (Fig. 2a). Experiments with both endogenous (Fig. 2a) and recombinant proteins (Fig. 3a, b) show that RNF8-mediated BLM polyubiquitylation is a determinant for this interaction. Hence BLM that cannot be polyubiquitylated (either lacking the three lysine residues or in cells where RNF8 is ablated) did not interact with NBS1 in vivo (Fig. 3c, d). Interestingly BLM interaction to MRN complex in vivo is via NBS1 and not MRE11 (Fig. 3e). These results together indicate that in vivo polyubiquitylated BLM's interaction with NBS1 leads to its further association with MRE11, allowing BLM to utilize the latter's exonuclease activity for its own recruitment. This

mechanism of BLM recruitment to DSBs is in contrast to the processes characterized for its localization to the stalled replication forks[5], where BLM recruitment predominantly depends RNF8/RNF168-mediated BLM ubiquitylation. Further these BLM ubiquitylation events differ from MIB1 and Fbw7α mediated polyubiquitylation of BLM, both of which lead to BLM degradation by 26S proteasome[32,33].

We have shown that depending on the phase of the cell cycle, BLM is co-recruited with HR and c-NHEJ factors, RAD51 and XRCC4, respectively (Fig. 4a–f, Supplementary Figure 10B-G). Perhaps more importantly BLM is crucial for the optimal recruitment of multiple HR and c-NHEJ factors to the chromatin in a cell cycle-specific manner (Fig. 4g–j, Supplementary Figure 12). The role of BLM in HR regulation has been well studied. BLM and RAD51 physically interact and colocalize at the sites of stalled replication[34–36]. Sumoylation of BLM regulates its interaction with RAD51[37]. We and others have previously shown that BLM disrupts RAD51 nucleoprotein filaments using both cell based and biochemical assays[14–16]. Interestingly we had also reported that the lack of BLM decreased the dynamic mobility and binding of RAD54 and RAD51 to the chromatin[28]. More recently evidence has been provided that BLM can regulate HR by counteracting RAD51 loading at the DSBs[38]. Thus our present data provide further mechanistic insight about how BLM regulates HR in S-phase.

To mechanistically understand how BLM has both a pro-recombinogenic role and also an anti-recombinogenic function, cells were grown in the presence of 4-OHT for 4 h and then for 1 h in media without any 4-OHT (schematic in Supplementary Figure 1B). This 1 h time period was vital as during this period the recruitment phase ceased to exist, the repair phase was initiated at the DSBs and consequently the anti-recombinogenic function of BLM became predominant. Hence while BLM activity was essential for the recruitment of HR and c-NHEJ factors in a cell cycle-specific manner during the recruitment phase (Fig. 4, Supplementary Figure 12), the same BLM activity inhibited HR and c-NHEJ (Fig. 5) in the repair phase due to the switch of BLM from a pro- to anti-recombinogenic role.

Finally, the present work provides evidence that BLM also has a negative regulatory role toward c-NHEJ, both in G1-and in S-phase. Thus in S-phase, BLM not only negative regulates HR but also c-NHEJ, thereby taking care of both the main and backup mechanisms of DNA repair (Fig. 5, Supplementary Figure 13, Supplementary Table 3). The fact that BLM is recruited equally to multiple DSBs irrespective of whether the DSB is in euchromatin or heterochromatin (Fig. 1b, Supplementary Figure 2D, E, 3) and can negatively regulate both HR and c-NHEJ in multiple phases

**Fig. 4** Cell cycle-dependent recruitment of HR and c-NHEJ factors is dependent on the helicase activity of BLM. **a–c** BLM, RAD51, and XRCC4 are co-recruited to the DSBs in S-phase. U2OS AsiSI were synchronized in S-phase (4 h post release from double thymidine block). Chromatin was prepared from these S-phase cells grown without (−4-OHT) or after 4 h of 4-OHT treatment. ChIP was carried out using **a** anti-BLM antibody; **b** anti-RAD51 antibody; **c** anti-XRCC4 antibody. BLM, RAD51, and XRCC4 recruitment to the indicated AsiSI-induced DSBs were determined by ChIP-qPCR. **d–f** BLM and XRCC4 are co-recruited to the DSBs in G1-phase. U2OS–AsiSI-ER cells were synchronized in G1-phase (18 h post release from double thymidine block). Chromatin was prepared from these G1-phase cells grown without (−4-OHT) or after 4 h of 4-OHT treatment. ChIP was carried out using **d** anti-BLM antibody; **e** anti-RAD51 antibody; **f** anti-XRCC4 antibody. Recruitment to the indicated AsiSI-induced DSBs or to the GAPDH loci were determined by ChIP-qPCR. **g**, **h** Helicase function of BLM is required for the cell cycle-dependent recruitment of RAD51 and XRCC4 to the DSBs. U2OS–AsiSI–ER cells were synchronized in either **g** S-phase or **h** G1 phase. Cells in either of the phases were subjected to 4 h of 4-OHT treatment, carried out either in the absence or presence of ML216. Parallel ChIPs were carried out with antibodies against RAD51 and XRCC4. The recruitment of RAD51 and XRCC4 to the indicated AsiSI-induced DSBs or to the GAPDH loci was determined by ChIP-qPCR. For all ChIP-qPCR analyses, the depicted values (mean ± standard deviation) were obtained from four independent experiments. Data were analyzed by unpaired two-tailed Student's t-test. **i**, **j** Helicase function of BLM is required for the cell cycle-dependent recruitment of HR and c-NHEJ factors to the chromatin. U2OS–AsiSI–ER cells were grown as in **g**, **h**. Soluble and chromatin fractions were isolated from cells and western analysis carried out with antibodies against BLM, HR factors (RAD51, RAD54, MRE11), c-NHEJ factors (Ku70, Ku80, Ligase IV, XRCC4), fractionation controls (hsp90 for soluble fraction and SMC1 for chromatin fraction). In each case, a Coomassie gel is included to demonstrate that equal amount of protein was taken for the analysis

of the cell cycle (Fig. 5, Supplementary Figure 13, Supplementary Table 3), validates the concept that that DSBs, irrespective of whether they are present in euchromatin or heterochromatin, can employ either c-NHEJ or HR for their repair[39].

In summary, we have determined that ATM, two members of the MRN complex and polyubiquitylation of BLM by RNF8 are

the key requirements for BLM recruitment to the DSBs. We have also provided evidence that BLM is the critical upstream component that is required for the recruitment of HR and c-NHEJ factors to the DSBs in a cell cycle-specific manner. We also show that during the repair phase BLM negatively regulates the repair processes, thereby switching from its initial role as a pro-repair protein to an anti-repair protein (Fig. 6). These two functions of BLM are essential to maintain the optimal level of DNA repair in the cells and thereby maintain genome integrity. Recently multiple mechanisms involving post-translation modifications of BLM have been deciphered. These include mechanisms which regulate the constitutive expression of BLM[40], its interaction with TopBP1 and thereby conferring stability[33], its turnover in both mitosis and G1-phase[32,33] and recruitment to PML-NBs[41]. It is quite possible that some these processes that regulate BLM cellular functions may act in close coordination with mechanisms by which BLM responds and ultimately repairs DSBs in different phases of the cell cycle.

## Methods

**Reagents**. The antibodies used are described in Supplementary Table 4. pcDNA3β myc-NBS1 (a gift of Xiaohua Wu), pcDNA3 myc-MRE11 (a gift of Xiaohua Wu), EGFP-C1 BLM (WT) and EGFP-C1 BLM (3K)[5], pGEX4T-1 BLM (1-1417)[28], mCherry2-C1 (a gift from Michael Davidson, Addgene plasmid #54563), NHEJ substrate pJS296 and I-SceI expression vector pJS20 (gifts from Jemery M. Stark). All siRNAs were purchased from Dharmacon and listed in Supplementary Table 5. Both siRNA transfections (using 200 pmol) and plasmid transfections (in HEK293T) were carried out for 48 h using Lipofectamine 2000 (Thermo Scientific).

**Cell culture**. U2OS–AsiSi–ER[21] (a gift of Gaelle Legube) and HCT116 BLM (−/−)[42] (a gift of Bert Vogelstein) cells were maintained as described. HEK293T cells were maintained in DMEM medium supplemented with 10% FCS. All cells used were tested free from mycoplasma contamination. Synchronization of U2OS–AsiSi–ER cells in S- and G1-phase by double thymidine block was carried out as published[32]. Supplementary Figure 1 summarizes the growth conditions of U2OS–AsiSi–ER cells used to study the role of BLM in either recruitment phase or

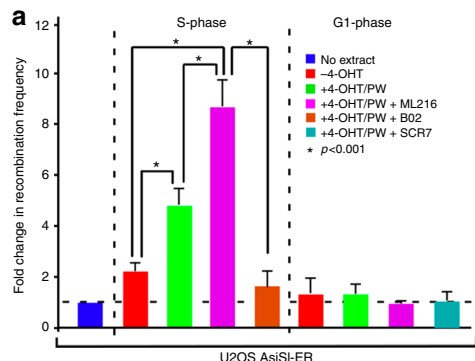

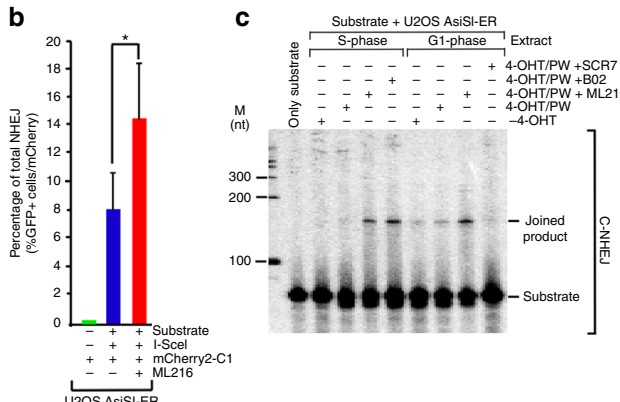

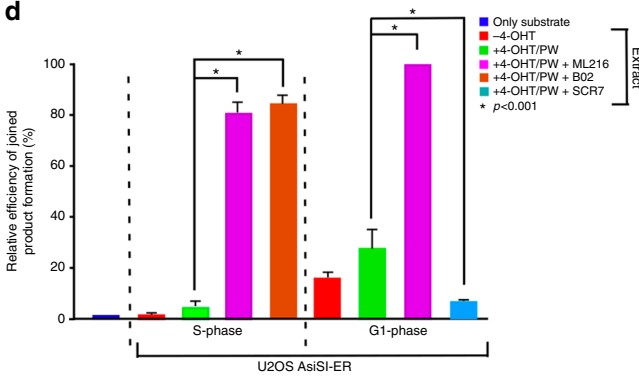

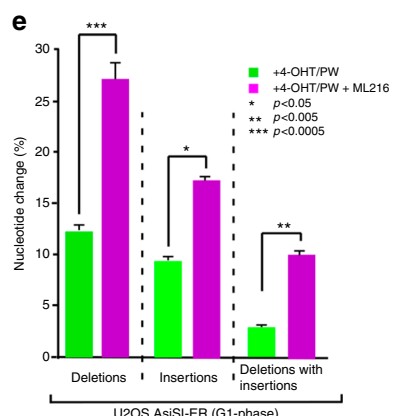

**Fig. 5** BLM negatively regulates both HR and c-NHEJ during repair phase. **a** Lack of functional BLM enhances HR in S-phase. HR assays were carried out using extracts from S- or G1-phase-arrested U2OS–AsiSI-ER cells grown in −4-OHT, +4-OHT/PW, +4-OHT/PW + ML216, +4-OHT/PW + B02, +4-OHT/PW + SCR7 conditions. Fold change in recombination frequency is represented with respect to the control where extract was not added. The entire experiment was done three times. Statistical analysis was done by Student's t-test. **b** Lack of functional BLM enhances total NHEJ using intrachromosomal substrates. NHEJ substrate (pJS296), I-SceI expression vector (pJS20) and mCherry2-C1 plasmids were co-transfected in asynchronously growing U2OS–AsiSI–ER cells grown in the absence or presence of ML216. The percentage of GFP-expressing cells was scored by FACS analysis. The experiment was done six times, data are presented as mean ± standard deviation and the statistical analysis was done by Student's t-test. **c**, **d** Lack of functional BLM enhances c-NHEJ in both S- and G1-phase. c-NHEJ assays were carried out using extracts, which were prepared as in **a**. **c** Representative autoradiogram is shown. **d** Percentage efficiency of joint product formation is represented. The product formed after ML216 treatment in G1-phase is taken as 100% and all other products are calculated relative to the above condition. Data are presented as mean ± standard deviation and the statistical analysis was done by Student's t-test. **e** Lack of functional BLM enhances nucleotide change at AsiSI sites. Genomic DNA was isolated from G1-phase-arrested U2OS AsiSI-ER cells grown in +4-OHT/PW, +4-OHT/PW + ML216 conditions. Thirty-five independent clones for each of the four AsiSI junctions were sequenced for both conditions. The percentage of nucleotide changes involving deletions, insertions and deletions with insertions are represented. Data were obtained from three independent experiments and represented as mean ± standard deviation, were analyzed by unpaired two-tailed Student's t-test

repair phase. In recruitment phase (Supplementary Figure 1A), U2OS–AsiSi–ER cells were either treated with vehicle (−4-OHT) or treated with 4-OHT (300 nM, Sigma). 4-OHT treatment was for a maximum of 4 h unless otherwise indicated. Cells were treated with Mirin (66 μM, Sigma) and KU55933 (12.9 nM, Sigma) for 1 h prior to and during the 4-OHT treatment. ML216 (12.5 μM, Sigma) treatment was carried out for 24 h prior to and during 4-OHT treatment. B02 (50 μM, Sigma) and SCR7 (66 μM) treatments were carried out during 4-OHT treatment for the last 3 h. For the repair phase (Supplementary Figure 1B), the extracts were made 1 h after 4-OHT was washed off during which the respective drugs used (ML216, B02, SCR7) continued to be present in the medium. Depending on the experiment, cells in −4-OHT conditions were treated with equal volume of the solvent in which ML216, Mirin, KU 55933, B02, or SCR7 were dissolved. Neocarzinostatin (NCS,

Sigma, 0.2 ng/μl) treatment of myc-tagged NBS1 and MRE11-transfected cells was for 1 h. The soluble nucleoplasmic (Fraction I) and chromatin bound (Fraction III) components were obtained as described[43]. For cell cycle analysis, the propidium iodide stained DNA content of the cells was sorted in a BD FACS Calibur and analyzed using the FlowJo software.

**Ubiquitylation, immunoprecipitation and immunofluorescence.** RNF8-mediated ubiquitylation of BLM[5] was carried out using 250 ng of the recombinant soluble substrate or 2 μl of in vitro transcribed and translated substrate (Promega). Assays were set up at 37 °C for 3 h in a total volume of 25 μl in 50 mM Tris-HCl (pH 8.0) and 1 mM DTT. Recombinant His-RNF8 (0.2 μM) was added to 0.4 μM of the indicated E2 enzymes Ubc13, 0.0125 μM E1 and 16 μM ubiquitin. Reactions were initiated by the addition of ATP (2 mM) and MgCl₂ (5 mM). The reactions were stopped by boiling with 2× SDS loading dye and samples were run on Nu-PAGE 4–12% gels in 1× MOPS buffer (Invitrogen), transferred onto a nitro-cellulose membrane and probed with the indicated antibodies. All lysates were made in RIPA buffer. Myc-tagged NBS1 and MRE11 were immunoprecipitated using 200 μg of RIPA extract and 1 μg of the antibody for 4 h. Interactions between immunoprecipitated myc-tagged NBS1 and MRE11 and BLM were carried out in 1× PBS + 0.1%NP40 buffer overnight at 4 °C. In vivo interactions between endogenous proteins were carried out by standard protocols using 500 μg of lysates and 2 μg of the antibody for 4 h. The protein–protein interactions with ChIP elutes was carried out after extracting the complex twice in elution buffer (0.1 M sodium bicarbonate, 1% SDS) for 15 min at 37 °C. The eluted fractions were pooled and processed for immunoprecipitations, as described above. For in vitro interactions, recombinant GST-BLM or GST (1 μg) was interacted with 5 μl of in vitro translated NBS1 carried out in presence of 20μCi S³⁵ methionine. The interaction was carried out at 4 °C for 4 h. Immunofluorescence followed by confocal microscopy was carried out using antibodies and their specific dilutions described in Supplementary Table 4. All incubations with primary antibodies were for 2 hrs while the corresponding secondary antibodies were incubated for 1 hr. Confocal imaging was carried out in LSM510 Meta System (Carl Zeiss, Germany) using 63x/1.4 oil immersion objective. The laser lines used were Argon 458/477/488/514 nm (For FITC) and DPSS 561 nm (for Texas Red). For immunofluorescence quantitation, a minimum of 200 cells across four biological replicates were analyzed. Data presented (mean ± standard deviation) were analyzed by unpaired two-tailed Student's t-test.

**ChIP assays.** ChIP assays using anti-BLM, anti-NBS1, anti-RAD51, and anti-XRCC4 antibodies were carried in U2OS–AsiSi–ER[21]. For this purpose, 250 μg of formaldehyde crosslinked chromatin was immunoprecipitated using 2 μg of the respective antibodies used for all the ChIP assays. After washing, the immuno-precipitated complexes were re-suspended and crosslinking reversed overnight. DNA was purified by phenol/chloroform, precipitated, and analyzed by ChIP-qPCR. Alternatively, ChIP was also carried out using the ChIP assay kit (Millipore, 17-295) as per the manufacturer's protocol. The extent of recruitment of the proteins at the AsiSI-generated cleavage sites was determined by ChIP-qPCR according to the percent input method. All the depicted values (mean ± standard deviation) were obtained from four independent experiments. Data were analyzed by unpaired two-tailed Student's t-test. The primers for ChIP-qPCR analysis are described in Supplementary Table 1, 2.

**Repair pathway assays and sequencing of AsiSI sites.** Lysates for HR and NHEJ assays were prepared[44] in a hypotonic lysis buffer (10 mM This-HCl pH 8, 1 mM EDTA, 5 mM DTT), followed by lysis by homogenization (20 strokes) in presence of protease cocktail inhibitors (Sigma) supplemented with individual

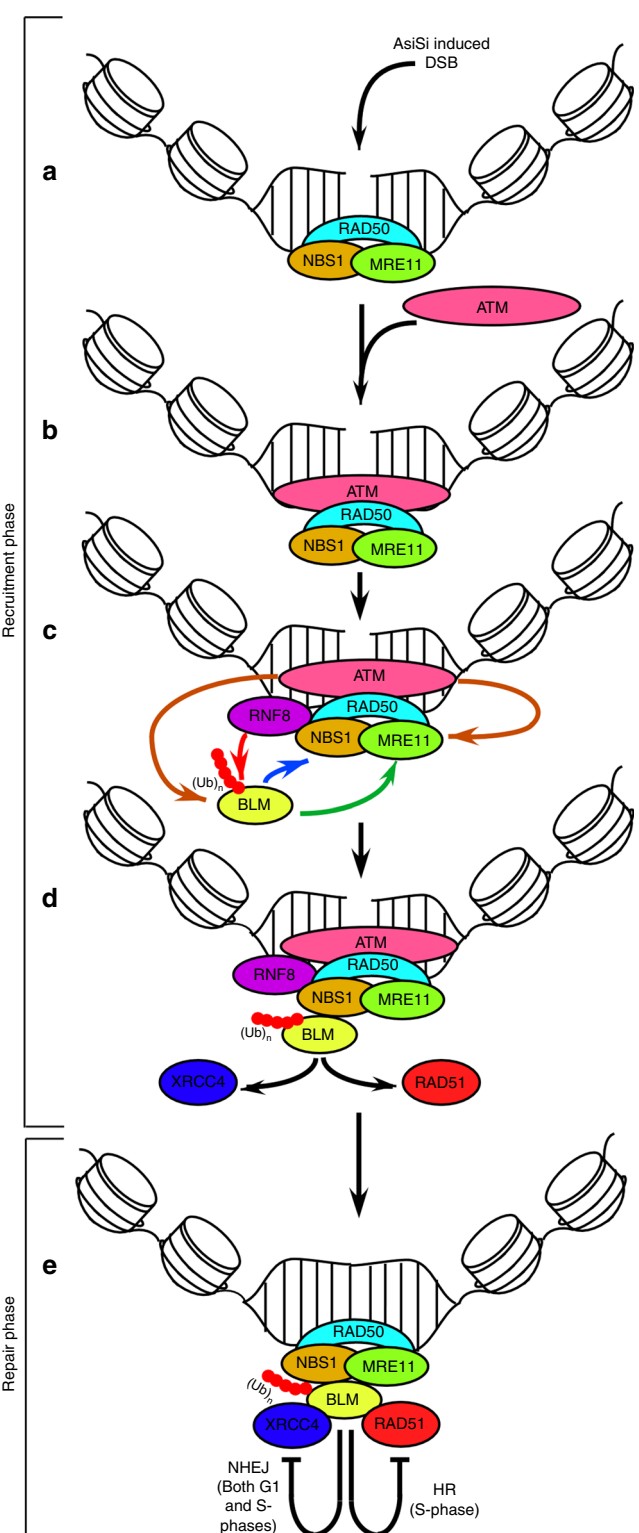

**Fig. 6** Mechanism of BLM recruitment to the DSBs and its effect on DNA repair. In the recruitment phase, MRN complex is recruited to the AsiSI-induced DSBs (**a**), which in turn recruits ATM (**b**). Chromatin-bound E3 ligase RNF8 polyubiquitylates BLM (red line, red dots representing ubiquitin residues). The kinase activity of ATM (brown lines) is essential for the BLM recruitment process, acting directly on BLM or via NBS1/MRE11. Single-stranded region obtained due to the exonuclease activity of MRE11 (green line) is also an essential requirement for BLM recruitment. Polyubiquitylated BLM interacts with NBS1 (blue line), which allows BLM to indirectly yet functionally interact with MRE11, leading to the optimal recruitment of BLM (**c**). The helicase activity of BLM is essential for the co-recruitment of c-NHEJ protein XRCC4 in G1-phase and RAD51 in S-phase (black lines) (**d**). During repair phase, BLM predominantly inhibits c-NHEJ in G1-phase and HR in S-phase. As a backup regulatory mechanism, BLM also inhibits c-NHEJ in S-phase (**e**). Hence BLM switches from a pro-repair protein (due to its role in the recruitment phase) to an anti-repair regulatory protein (due to its role in the repair phase), thereby maintaining genomic stability

components. The samples were kept on ice for 20 min, 0.5 volume of high salt buffer (50 mM Tris-HCl, pH 7.5, 1 M KCl, 2 mM EDTA, 2 mM DTT) added. The lysates were ultracentrifuged for 3 h at 211,422.6$g$ in a Beckman SW50.1 rotor. Post centrifugation, the sample was dialyzed for 3 h against dialysis buffer (20 mM Tris-HCl, pH 8.0, 0,1 KOAC, 20% glycerol v/v, 0.5 mM EDTA, 1 mM DTT) and frozen in deep freezer. In vitro HR assays were carried[45,46] using (500 ng) of the substrates (pTO231 and pTO223 both of which have ampicillin resistance). Reciprocal recombination and gene conversion between the two substrates lead to the generation of a functional neomycin gene that conferred kanamycin resistance. In the assays, the substrates were incubated with 5 μg of the cell extracts prepared under different conditions. The reaction was carried at 37 °C for 30 min in an HR reaction buffer (35 mM HEPES-pH-8.0, 10 mM MgCl$_2$, 1 mM DTT, 2 mM ATP, 50 μM dNTPs, 1 mM NAD, 100 μg/ml BSA). DNA obtained after Proteinase K treatment, phenol–chloroform extraction, and ethanol precipitation was dissolved in TE buffer and plated in 1:10 ratio between Luria Broth plates containing either ampicillin or kanamycin. Colonies counted after electroporation was used to calculate fold change in recombination frequency. The entire experiment was done three times. Statistical analysis was done by Student's $t$-test. Standard methods of SCE determination were followed[35]. Forty metaphase spreads were analyzed for each condition. Three independent experiments were carried out. The data are presented as mean ± standard deviation and the $p$-values were obtained by Student's $t$-test. The conditions are two-tailed, unpaired data with unequal variance.

NHEJ assays with intrachromosomal substrates were carried out six times[24,47] with certain modifications. NHEJ substrate pJS296 (1 μg) and I-SceI expression vector pJS20 (2 μg) were transfected in asynchronously growing U2OS–AsiSI–ER cells (−4-OHT) in the absence or presence of ML216. mCherry2-C1 plasmid (0.5 μg) was transfected in each case to determine the transfection efficiency. Thirty-six hours post transfection, the cells were collected in PBS and 50 mM EDTA, pelleted and fixed with 2% paraformaldehyde for 20 min. The percentage of mCherry and GFP-expressing cells was scored by FACS analysis in BD FACS AriaIII. The experiment was done six times, data are presented as mean ± standard deviation and the statistical analysis was done by Student's $t$-test. From parallel plates, lysates were made and transfection efficiency was also determined by western blot analysis using an anti I-SceI antibody.

In vitro c-NHEJ assays were carried[24] with [γ-32P] ATP end-labeled double-stranded oligonucleotides containing 5′ compatible ends (top strand: 5′ GAT CCC TCT AGA TAT CGG GCC CTC GAT CCG GTA CTA CTC GAG CCG GCT AGC TTC GAT GCT GCA GTC TAG CCT GAG 3′; bottom strand: 5′ GAT CCT CAG GCT AGA CTG CAG CAT CGA AGC TAG CCG GCT CGA GTA GTA CCG GAT CGA GGG CCC GAT ATC TAG AGG 3′). The data presented are from three biological replicates and are represented as mean ± standard deviation. Data were analyzed by Student's $t$-test.

The genomic DNA around AsiSI sites was cloned in pUC18 and subjected to Sanger sequencing. In both −ML216 and +ML216 condition, 140 individual clones, spanning across four AsiSI sites obtained from three independent experiments were sequenced. Fifty to one hundred nucleotides on both sides of the AsiSI sites were considered to determine the accuracy of the end joining. Data were obtained from three independent experiments and represented as mean ± standard deviation, and were analyzed by unpaired two-tailed Student's $t$-test. The raw sequences from which the data were extracted to determine the nucleotide alterations at or near to the AsiSI sites have been submitted in FigShare: https://figshare.com/articles/Chr6_AsiIS_90404906_pdf/5852853.

**Data availability**. The individual source files of all the western blots and autoradiograms, presented as TIFF images, are attached as supplementary datasets. The combined source file for all the data are presented in Supplementary Figure 14. Individual source files have been submitted in FigShare: https://figshare.com/articles/Source_File_for_Images/5852940. All other relevant data are available from the authors.

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

## Acknowledgements

We acknowledge Xiaohua Wu and Jemery M. Stark for plasmids, Gaelle Legube and Bert Vogelstein for cells, and Sumedha Dahal for HR assays, Preeti Attri for FACS analysis, Next Generation Sequencing Facility, NII for help during in silico analysis. S.S. acknowledges the National Institute of Immunology core funds, Indo-French Centre for the Promotion of Advanced Research (IFCPAR) (IFC/4603-A/2011/1250), Council of Scientific and Industrial Research (CSIR) [37(1699)/17/EMR-II], Department of Bio-technology (DBT), India (BT/PR7320/BRB/10/1161/2012 and BT/MED/30/SP11263/2015), and Science and Engineering Research Board (SERB), India (SR/SO/BB-0124/2013) for financial assistance. V.T. acknowledges the Department of Biotechnology for a Research Associate (DBT-RA) position.

## Author contributions

V.T., H.A., S.P., H.B., P.M., M.P., and D.S. carried out the experiments and analyzed the data. S.C.R. designed HR, c-NHEJ experiments, and analyzed the data. S.S. designed the overall experiments, analyzed the data, and wrote the manuscript.

## Additional information

**Competing interests:** The authors declare no competing interests.

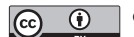

