## [Peer Review File · Nature Communications]

Reviewers' comments:

Reviewer #1 (Remarks to the Author):

This paper evaluates the requirements for recruitment of BLM to DSBs on a quasi-global basis and as a function of the cell cycle using AsiSI to introduce DSBs. The authors find BLM recruited to DSBs, identified as H2AX foci. They show recruitment is independent of BLM helicase activity but dependent on presence of Mre11, Nbs1 and ATM, in both G1 and S phase. In addition, they provide evidence that Mre11 nuclease but not ATM kinase activity is required. Interestingly, BLM ubiquitylation by RNF8 is also required for full recruitment. Only after simultaneous inhibition of Mre11 nuclease and depletion of RNF8, is recruitment abolished. BLM recruitment and active BLM helicase are required for RAD51 recruitment in S phase and for XRCC4 in G1 phase. Rad51 is not recruited in G1. Another interesting finding is that after induction of DSBs there is a biphasic recruitment of BLM, an early phase just after induction of the restriction enzyme causing the breaks and then a decrease and then a larger increase after 4 h. Most of the studies relate to the breaks that exist after 4 h. Finally, they show that inhibition of BLM helicase after DSB induction increases HR in S phase by an *in vitro* recombination assay. This revealed a role for BLM in cNHEJ joining assays, since BLM inhibitors stimulated cNHEJ after DSB induction in G1 (as well as in S phase) and this correlated with an increase in mutagenesis. These results are worth publishing and of interest because recruitment of BLM to stalled replication forks has been well studied but recruitment to specific DSBs, allowing detailed studies, and combination with the use of recently described inhibitors instead of knockdowns, provides a substantial improvement in our understanding of how BLM regulates pathway choice at DSBs.

Most of the experiments are well done and data are clean. A few improvements are suggested:

Fig. 1: An independent assay of the time course of induction of DSBs by AsiSI would be helpful.

Also, was a chromosomal locus lacking a DSB studied as a control?

Fig. 2 There is probably a typo in the legend to Fig 2 B,C. The data and text say that BLM recruitment is dependent on the presence of Mre11, Nbs1 and ATM but the legend says "independent of."

Fig. 4 Again there is a confusing typo, Fig. 4 G-J are labeled Fig. 5G-J in the text. I couldn't find the data labeled as Fig. S6A and B that seemed relevant (p. 8, para. 2).

Fig. 5. The HR recombination assays need to be described briefly for the reader, who will not want to go back to the original papers. While the results show clear differences in HR it would be nice to see some examples of the data, as shown for NHEJ in B, to be sure they support the interpretations in the text, it is hard to evaluate the results without understanding the assays and how they were originally validated as reflecting *in vivo* functions.

Reviewer #2 (Remarks to the Author):

BLM helicase plays an important role in recognition and resolution of DNA damage in replicative stress conditions. Mutation in BLM leads to cancer predisposition. Its recruitment during replication stress, an important source of genome instability, has been deeply studied. It is also important to well characterize its recruitment to double-strand breaks, to also elucidate its pro- or anti-repair function.

In this paper, Tripathi et al used the Diva system (an inducible site-specific double-strand breaks generating system) to analysed the recruitment of BLM on DSB. They show that (1) BLM is recruited from 80 bps to 9kb from the break site ; (2) BLM recruitment is independent of its own helicase activity but it is dependent of NBS1, MRE11 and ATM (as it has been previously reported) ; (3) BLM recruitment is abrogated upon MRE11 exonuclease activity inhibition and inhibition of RNF8 (which mediates the ubiquitination of BLM protein), (4) BLM is co-recruited with RAD51 in S-phase (as it has been reported upon replicative stress), and with XRCC4 in G1 phase, its helicase activity is important for the recruitment of key HR and c-NHEJ factors onto chromatin.

The authors propose that BLM inhibits HR in S-phase and c-NHEJ in G1 phase. The authors show also that ubiquitylated BLM interacts with NBS1, this latter data represents a novelty.

Although this study presents interesting data, concerns exist and some apparent discrepancy exist with data published previously which should be addressed or discussed before publication. This paper should be published somewhere, but although some data are original, the novelty of the concepts are still questionable, and may be not sufficient to be published in Nature communications.

1- Page 3, the authors claim that «nothing is known how BLM is recruited to another more prevalent, potent and physiological class of lesions, the DSBs. » This is a little exaggerated, they exist some data in the literature.

2- « BLM has also been implicated in the error-prone MMEJ. Hence, in cells lacking BLM, the rate of microhomology-mediated rearrangements was enhanced ». These two sentences seem contradictory. In the 3 references cited, it is shown that a deficiency of BLM leads to deletions or rearrangements mediated by the MMEJ. Hence, BLM prevents this inaccurate pathway of DSBR.

3- "DSB induction caused BLM recruitment to all the tested ten DSBs spreads across nine chromosomes". 7 different DSBs are shown.

4- BLM is recruited to the DSBs in a biphasic manner. This observation is important. It would be important to determine what this second wave of recruitment is. This is particularly important here because most experiments were performed after 4 hours OH-TAM which corresponds to the second wave and not the initial recruitment during the first wave. Some key ChIP experiments to characterize the factors required for BLM recruitment should be also performed at 30min after 4-OH-TAM.

Would this second wave be observed after ionizing radiation (1 or 2 Gy)?

5- Data showing that NBS1, but also ATM-dependent phosphorylation of CtIP is important for BLM recruitment to DNA damages has been previously published (Wang et al, Plos Genet, 2013). The apparent discrepancy with the present data proposing that ATM kinase is not required for BLM recruitment should be addressed or discussed.

The impact of ATM activity inhibitor has to be performed 30 min after 4-OHT. The impact of the phosphorylation of BLM (Thr99) by ATM on recruitment (by ChIP) after 30 min and 4 hours of 4OH-TAM could be tested.

6- in Figure S4 C-D -pThr68Chk2 is observed even without induction of AsiSI. Why Mirin or KU55933 inhibitors do not decrease the level of p-Chk2 in these basal conditions?

7- Page 7 : « Ubiquitylated BLM interacts with NBS1 at the DSBs ». This paragraph title has to be changed. There is no direct evidence that these interaction are localized at DNA damage. Only co-IP were performed after NCS treatment.

« Direct interaction of ubiquitylated BLM occurs with NBS1... »

It is not possible to talk about "direct" interaction since this co-iP was performed using immunoprecipitated NBS1 from human cellular extracts.

Fig 3- Co-IPs was performed after overexpression of Myc tagged NBS1. It will be interesting to perform co-IPs with endogenous proteins.

Fig. 3C: Ubiquitylated BLM does not interact with immunoprecipitated Mre11. Would endogenous NBS1 also not be present in the immunoprecipitate? No trace of ubiquitylated BLM is visible, is this really expected?

It will be informative to perform the co-IPs in absence of NCS treatment.

The ubiquitination sites of BLM by RNF8 are known. It would be interesting to check whether the mutant BLM 3K (previously described) does not interact with endogenous NBS1. (Co-IPs experiments may be performed using BLM-deficient cells transfected with BLM wt or mutant BLM 3K.)

Fig 3D, E : Depletion of RNF8 affects the recruitment of BLM to DSB induced by AsiSI. However, since RNF8 has numerous targets, essential for DSB repair. This observation not allow to determine if direct ubiquitination of BLM by RNF8 is required for the recruitment of BLM to DSBs.

Protocol should more detailed, for instance Fig 3D, have the conditions without Mirin been carried out in the presence of DMSO (vehicle)?

8) Fig S5C-S5E has to be change for Fig S6A-E. The supplemental figures are not well referenced.

In G1, ML216 inhibitor decrease XRCC4 in chromatin fraction (Fig4J). How to reconcile this data with the fact that this treatment increases the rate of joined product formation?

The author claim that « BLM negatively regulated c-NHEJ in this phase ».

Depletion of a key factor of canonical NHEJ, such as lig 4, or SCR7 treatment has to be perform in combination with ML216 to confirm this asumption. In addition, the impact of BLM on DSBR pathways could be performed using intrachromosomal substrates. This type of substrates are more relevant.

Some data would be also performed BS cells. Are recruitment of these HR and NHEJ factors affected in BS cells?

Moreover, the authors show that after 4-OHT, absence of BLM increased rates of sequences alterations, « providing BLM helicase functions affects c-NHEJ in G1 phase ». Overall, these data seem contradictory and interpretations quite confusing.

However, different studies (in vitro using BS cells extracts, or by ligation of episomal plasmids upon BLM depletion, and even more recently by using intrachromosomal substrates) showing that BLM deficiency leads to a decrease of fidelity of end-joining have been already published (Games, Oncogene, 2002 ; Grabarz, Cell report, 2013...). Thus, results described in Table S3 are not really original.

Anyway, number of sequenced clones in this study is not sufficient (15 for each conditions, comparison between 0, 1 or 2 clones on 15 total clones for insertions for instance is not really relevant !).

9. Discussion-

Page 11 « Co-depletion experiments show that the binding of BLM to the single-stranded DNA causes by... »

should be replace by « Co-depletion experiments suggest.. » Here there is no data on direct binding of BLM on ssDNA, its is only suggested by the requirement of Mirin for its recruitment, thus potentially also the requirement of resection. Its binding could be through another factor on ssDNA.

« This mechanism of BLM recruitment to DSBs is in contrast to the process characterized for its localiation to stalled forks ». It would be interesting to summarize what is clearly different.

Anti-recombination function of BLM is mediated by poly-ubiquitination (Böhm, 2014) has been reported. Here the autors show that poly-ubiquitylation of BLM is required for binding of NBS1. The data should be discussed in relation to what is known in the literature.

« Expectedly BLM inhibits only c-NHEJ in G1 phase... » It has been reported that BLM counteracts MMEJ. This apparent discrepancy should be discussed.

Answers to reviewer's comments

Reviewer #1:

Comment #1:

Fig. 1: An independent assay of the time course of induction of DSBs by AsiSI would be helpful. Also, was a chromosomal locus lacking a DSB studied as a control?

Answer:

We appreciate the comments made by the reviewer. In the revised version there are two independent assays which show increased induction of DSBs by AsiSI over time. While Figure S2A depicts the levels of γ H2AX over a time period using western blot as a technique, Figure S2C shows the extent of BLM/ γ H2AX foci colocalization after 0.5 hr and 4 hrs post-DSB induction due to 4-OHT treatment. Both these assays have been indicated in Page 5, top paragraph of the revised manuscript.

Yes – throughout the manuscript the GAPDH loci which lacks any AsiSI induced DSB has been used as a control in the ChIP-qPCR experiments to show the specificity of the recruitment of the DDR proteins.

Comment #2:

Fig. 2 There is probably a typo in the legend to Fig 2 B,C. The data and text say that BLM recruitment is dependent on the presence of Mre11, Nbs1 and ATM but the legend says “independent of.”

Answer:

We appreciate the mistake pointed out by the reviewer. We have corrected the mistake in the legend of Figure 2B, 2C (page 28 of the revised version). BLM recruitment to the annotated DSBs is indeed dependent on Mre11, Nbs1 and ATM.

Comment #3:

Fig. 4 Again there is a confusing typo, Fig. 4 G-J are labeled Fig. 5G-J in the text. I couldn't find the data labeled as Fig. S6A and B that seemed relevant (p. 8, para. 2).

Answer:

We completely agree with the reviewer's comments. Indeed there was wrong labelling of the figures in Page 9, 2nd paragraph. We have now corrected the labelling as Figure 4G, 4H, 4I, 4J in the revised version (page 9). Figure S6A and S6B are actually Figure S12A, S12B in the revised version. All the corrected figure numbers are highlighted in Page 9, 2nd paragraph.

Comment #3:

Fig. 5. The HR recombination assays need to be described briefly for the reader, who will not want to go back to the original papers. While the results show clear differences in HR it would nice to see some examples of the data, as shown for NHEJ in B, to be sure they support the interpretations in the text, it is hard to evaluate the results without understanding the assays and how they were originally validated as reflecting *in vivo* functions.

Answer:

We appreciate the reviewer's concern. We have included the protocol of the HR assay in some detail along with the guiding principle of the assay (page 17 of the revised manuscript). HR frequency (i.e. formation of a functional neomycin gene) is measured by the relative number of bacterial colonies obtained in the kanamycin and ampicillin plates after electroporation of the transformation mix. Since transformation and counting bacterial colonies is a routine assay, pictorial representation of the colonies has not been incorporated.

Reviewer #2:

Comment #1:

Page 3, the authors claim that “nothing is known how BLM is recruited to another more prevalent, potent and physiological class of lesions, the DSBs.” This is a little exaggerated, they exist some data in the literature.

Answer:

We agree with the reviewer’s comments. In the revised version (page 3, 2nd paragraph) we have now incorporated all the four known references regarding BLM recruitment to the DSBs.

Comment #2:

“BLM has also been implicated in the error-prone MMEJ. Hence, in cells lacking BLM, the rate of microhomology-mediated rearrangements was enhanced”. These two sentences seem contradictory. In the 3 references cited, it is shown that a deficiency of BLM leads to deletions or rearrangements mediated by the MMEJ. Hence, BLM prevents this inaccurate pathway of DSBR.

Answer:

We agree that in the earlier version of the manuscript we were not clear about BLM’s role with respect to MMEJ. BLM is known to negatively regulate MMEJ and this has been clearly indicated in the revised version (Page 3, last paragraph and Page 4, top paragraph).

Comment #3:

“DSB induction caused BLM recruitment to all the tested ten DSBs spreads across nine chromosomes”. 7 different DSBs are shown.

Answer:

We thank the reviewer for pointing it out. Though we have done BLM ChIP for ten DSBs, we have presented the data for seven DSBs in the mentioned figures. We have corrected this mistake in the revised version (page 5, 2nd paragraph).

Comment #4A:

BLM is recruited to the DSBs in a biphasic manner. This observation is important. It would be important to determine what this second wave of recruitment is. This is particularly important here because most experiments were performed after 4 hours OH-TAM which corresponds to the second wave and not the initial recruitment during the first wave. Some key CHIP experiments to characterize the factors required for BLM recruitment should be also performed at 30min after 4-OH-TAM.

Answer:

To address the comments we have carried out three key experiments:

- a. Carried out parallel ChIPs with anti-BLM, anti-NBS1, anti-XRCC4 and anti-RAD51 antibodies after exposure of U2OS AsiSI-ER cells to 4-OHT for 30 minutes. By this experiment we wanted to determine whether BLM gets co-recruited with NBS1 (a known DNA damage sensor protein) and/or with XRCC4/RAD51 (NHEJ and HR proteins) in the first wave of recruitment. We found that in this early phase of recruitment BLM was co-recruited with NBS1 but not appreciably with RAD51 and XRCC4. These results are presented in Figure 1E, S4B (page 6, 3rd paragraph).
- b. Secondly we also wanted to determine whether BLM could be recruited to the DSBs in absence of ATM, MRE11 and NBS1 when the U2OS AsiSI-ER cells were exposed to 4-OHT for either 30 minutes or 4 hrs. For this purpose, parallel ChIP-qPCRs was carried out in cells transfected with either siControl or siNBS1 or siMRE11 or siATM. These cells were treated with 4-OHT for either 0.5 hr or 4 hrs. The results presented in Figure 2B, 2C, S6 and S7 show that at both time points BLM recruitment was dependent on ATM, MRE11 and NBS1 (page 6 last paragraph and page 7 first paragraph).
- c. Finally we wanted to determine whether BLM is recruited to the DSBs in absence of either MRE11 exonuclease activity (due to Mirin treatment) or ATM kinase activity (due to KU 55933 treatment) after exposure to either 0.5 hr or 4 hrs of 4-OHT treatment. We found that at both 0.5 hr and 4 hrs, lack of MRE11 exonuclease activity inhibits optimal BLM recruitment (Figure 2D, S8C). Interestingly the lack of the kinase activity of ATM is essential for optimal BLM recruitment only in the early phase (i.e. 0.5 hr). In the late phase (4 hrs post-DSB generation) kinase activity of ATM is not required for BLM recruitment to the DSBs (Figure 2E, S8D). These datasets are presented in page 7, 2nd paragraph of the revised version.

Comment #4B:

Would this second wave be observed after ionizing radiation (1 or 2 Gy)?

Answer:

Since AsiSI induces clean DSBs it is expected that these results regarding BLM recruitment would also apply to IR exposure. However experimentally this cannot not be tested out easily as IR would induce DSBs in a random manner all over the genome and not at specific annotated sites. Hence ChIP-qPCRs cannot be done after exposing cells to IR. However in theory ChIP-seq can be attempted to answer this question – which we consider to be outside the scope of the present manuscript. However the broad question is something the lab is currently working on and hopefully we should know in future how BLM responds to low and high doses of IR.

Comment #5A:

Data showing that NBS1, but also ATM-dependent phosphorylation of CtIP is important for BLM recruitment to DNA damages has been previously published (Wang et al, PloS Genet, 2013). The apparent discrepancy with the present data proposing that ATM kinase is not required for BLM recruitment should be addressed or discussed. The impact of ATM activity inhibitor has to be performed 30 min after 4-OHT.

Answer:

It is indeed true that it has been reported that ATM-dependent phosphorylation of CtIP is important for BLM recruitment to the sites of laser induced DSBs (Wang et al, Plos Genet, 2013). In this publication it has been shown that in cells expressing CtIP mutants which cannot be phosphorylated by ATM, the recruitment of BLM is compromised upto 12 minutes. We also see a similar phenomenon when we carry out ChIP-qPCR in U2OS AsiSI-ER cells after 0.5 hr of exposure to 4-OHT. We find that under this condition the recruitment of BLM to the DSBs is compromised (Figure 2E). Hence our data is in concordance with the published literature.

Comment #5B:

The impact of the phosphorylation of BLM (Thr99) by ATM on recruitment (by ChIP) after 30 min and 4 hours of 4OH-TAM could be tested.

Answer:

It should have been interesting to carry out ChIP using anti-pThr99BLM antibody. We indeed tried to standardize ChIP using anti-pThr99BLM antibody (ab62206, Abcam). Unfortunately ChIP using this antibody did not work.

Comment #6:

In Figure S4 C-D -pThr68Chk2 is observed even without induction of AsiSI. Why Mirin or KU55933 inhibitors do not decrease the level of p-Chk2 in these basal conditions?

Answer:

We do not know why pThr68Chk2 was seen even without induction of DSBs by AsiSI treatment (Figure S5C, S5D of the previous version of the manuscript). We can only speculate that the pThr68Chk2 signal under -4-OHT condition (without and with Mirin or KU 55933 treatment) was a measure of the endogenous DNA damage in U2OS AsiSI-ER cells which activated Chk2 in a non-cannonical manner (i.e. not via MRN/ATM/Chk2 axis). However it is to be also noted that other phospho-antibodies of the DDR proteins (namely pSer1981ATM, pSer15p53, pThr99BLM) acted more conventionally. While using these three antibodies a band was visualized in asynchronous conditions (possibly indicating basal DNA damage), which either disappeared or was reduced after Mirin or KU 55933 treatment. Moreover, we found that in past doubts have been raised regarding the specificity of the commercially available pThr68Chk2 antibodies especially with respect to foci formation after DSB generation [Bartek and Lukas, Cancer Cell, 3: 421-429 (2003)]. Keeping the above arguments in mind we have removed the Chk2 and pThr68Chk2 data from the revised manuscript (Figure S8A and S8B in the revised manuscript). However other blots involving activated and non-activated forms of BLM, ATM and p53 show that both Mirin and KU 55933 were active during the experiments.

Comment #7A:

Page 7: “Ubiquitylated BLM interacts with NBS1 at the DSBs”. This paragraph title has to be changed. There is no direct evidence that these interaction are localized at DNA damage. Only co-IP were performed after NCS treatment.

Answer:

We agree with the reviewer’s reasoning. Taking reviewer’s concern into consideration we have modified the heading in Page 7 as “Polyubiquitylated BLM and NBS1 are components of the same complex after DSB generation”.

Comment #7B:

“Direct interaction of ubiquitylated BLM occurs with NBS1...”
It is not possible to talk about "direct" interaction since this co-iP was performed using immuno-precipitated NBS1 from human cellular extracts.

Answer:

To answer this query we have carried out in vitro interaction between GST-BLM and in vitro translated NBS1. A low affinity direct physical interaction between BLM and NBS1 was detected (Figure S9, page 7, bottom paragraph). However the interaction between NBS1 and BLM was greatly enhanced when BLM was polyubiquitylated in a RNF8 dependent manner (Figure 3A, 3B, 3E).

Comment #8A:

Fig 3- Co-IPs was performed after overexpression of Myc tagged NBS1. It will be interesting to perform co-IPs with endogenous proteins.

Answer:

We have indeed done Co-IPs with endogenous proteins using lysates from U2OS AsiSi-ER cells (\pm 4-OHT conditions). It is presented as Figure 2A of the revised manuscript.

Comment #8B:

Fig. 3C: Ubiquitylated BLM does not interact with immunoprecipitated Mre11. Would endogenous NBS1 also not be present in the immunoprecipitate? No trace of ubiquitylated BLM is visible, is this really expected?

Answer:

In this experiment (Figure 3E of the revised version) the interaction was carried out between polyubiquitylated or non-polyubiquitylated BLM and Myc-tagged NBS1 or Myc-tagged MRE11. After interaction, the blots were probed with anti-BLM antibody. Due to the overexpression based system, it is not known how much of functional MRN complex is formed when only one component (either Myc-tagged NBS1 or Myc-tagged MRE11) is overexpressed, because of the improper stoichiometry of the interacting partners. This is probably the reason why polyubiquitylated BLM is not visible when Myc-tagged MRE11 is made to interact with polyubiquitylated BLM. However we agree that in an ideal experiment a bit of polyubiquitylated BLM should also have been visible when the above interaction was carried out.

Comment #8A:

It will be informative to perform the co-IPs in absence of NCS treatment.

Answer:

We appreciate the reviewer's viewpoint. In response we have carried out interactions between in vitro ubiquitylated and non-ubiquitylated BLM with NBS1 obtained from cells which has not been exposed to NCS treatment. We found that if BLM is polyubiquitylated in vitro by RNF8 it will interact with NBS1 irrespective of when the cells have been exposed to NCS or growing asynchronously. This has been incorporated as Figure 3A of the revised version.

In a complementary experiment Co-IPs were performed in U2OS AsiSI-ER cells with or without 4-OHT treatment (Figure 2A). While BLM interacted with NBS1 in absence of DNA damage, the interaction was greatly enhanced after 4-OHT treatment.

Comment #8B:

The ubiquitination sites of BLM by RNF8 are known. It would be interesting to check whether the mutant BLM 3K (previously described) does not interact with endogenous NBS1. (Co-IPs experiments may be performed using BLM-deficient cells transfected with BLM wt or mutant BLM 3K.)

Answer:

We have responded to reviewer's comment by carrying out transfection of BLM (WT) or BLM (3K) in HCT116 BLM (-/-) cells, exposing the cells to NCS before preparing the lysates. Immunoprecipitation of endogenous NBS1 revealed that BLM (WT) but not BLM (3K) interacted with endogenous NBS1. This experiment has been incorporated in Figure 3C in the revised version.

Comment #9:

Fig 3D, E : Depletion of RNF8 affects the recruitment of BLM to DSB induced by AsiSI. However, since RNF8 has numerous targets, essential for DSB repair. This observation not allow to determine if direct ubiquitination of BLM by RNF8 is required for the recruitment of BLM to DSBs.

Answer:

The reviewer has observed that Figure 3F, 3G, 3H of the revised version (which correspond to Figure 3D, 3E of the first version) does not necessarily say that direct ubiquitination of BLM by RNF8 is required for the recruitment of BLM to DSBs. We have checked BLM recruitment to the DSBs by two methods – BLM ChIP (Figure 3F) and BLM foci formation (Figure 3G, 3H) – both of which are bonafide techniques

to determine recruitment of proteins onto the chromatin. However we acknowledge that formally it maybe possible that one of the many other RNF8 targets maybe involved in BLM recruitment. Hence we have written “However, depletion of both ubiquitylation and MRE11 exonuclease activities completely abolished BLM recruitment to the DSBs (Figure 3F-3H), **suggesting** that RNF8 mediated ubiquitylation along with MRE11 exonuclease activity are key events which allows BLM to be recruited to the chromatin” (page 8, 2nd paragraph).

Comment #10:

Protocol should more detailed, for instance Fig 3D, have the conditions without Mirin been carried out in the presence of DMSO (vehicle)?

Answer:

We have added more details to all the experimental methods. Specifically for this query we have stated “Depending on the experiment, cells in -4-OHT conditions were treated with equal volume of the solvent in which ML216, Mirin, KU 55933, B02 or SCR7 were dissolved” (page 15, bottom). This has also been stated in the legend of Figure S1.

Comment #11:

Fig S5C-S5E has to be change for Fig S6A-E. The supplemental figures are not well referenced.

Answer:

We thank the reviewer for pointing out the mistake. Figure S5C-S5E of the previous version is now re-designated as Figure S11B-S11E in the present version (page 9).

Comment #12A:

In G1, ML216 inhibitor decrease XRCC4 in chromatin fraction (Fig4J). How to reconcile this data with the fact that this treatment increases the rate of joined product formation?

Answer:

Figure 4G-4I demonstrates the effect of BLM helicase activity on the recruitment of the HR and NHEJ factors. On the other hand, Figure 5A-5E tells about the effect of

BLM helicase activity on the repair processes, HR and NHEJ. To mechanistically distinguish between the two processes – we have differentiated between the two experimental conditions. In recruitment phase (for experiments done in Figure 4) the U2OS ASiSI-ER cells are continuously exposed to 4-OHT for 4 hrs. However in the repair phase (for experiments done in Figure 5A-5E), U2OS AsiSI-ER cells after first exposed to 4-OHT for 4 hrs, after which the 4-OHT is washed off and the repair is allowed to proceed (in presence of ML216, B02, SCR7) for 1 hr. We have specifically explained the change in the experimental conditions both in the text (page 9, bottom paragraph and page 15 bottom paragraph) and also in Figure S1.

In the recruitment phase BLM acts as a pro-repair protein as the absence of its helicase activity prevents the recruitment of the repair factors (Figure 4G-4J). However in the repair phase BLM acts as an anti-repair protein for both HR and NHEJ and thereby negatively regulates both HR and c-NHEJ (Figure 5A-5E). Hence we believe that we have been able to experimentally dissect and demonstrate the two antagonistic functions of BLM during DDR. It is to be noted that based on in vitro results both the pro-and anti-repair functions of BLM had been postulated earlier [Bugreev et al., *Genes and Dev*, 21, 3085-3094 (2007)]. This is the first in vivo validation of these two BLM functions with respect to DSBs.

Comment #12B:

The author claim that “BLM negatively regulated c-NHEJ in this phase”. Depletion of a key factor of canonical NHEJ, such as lig 4, or SCR7 treatment has to be performed in combination with ML216 to confirm this assumption.

Answer:

We suggested by the reviewer we have carried out the in vitro c-NHEJ assay (i.e. the joined product formation) in presence of both SCR7 and ML216. The presence of both ML216 and SRC7 completely abrogated c-NHEJ in G1 phase, further validating that BLM negatively regulates c-NHEJ (Figure S13B, S13C, page 10 first paragraph).

Comment #12C:

In addition, the impact of BLM on DSBR pathways could be performed using intrachromosomal substrates. This type of substrates are more relevant.

Answer:

We agree with the reviewer’s suggestion. Hence we have carried out NHEJ assays with intrachromosomal substrate, pJS296 [Bennardo, *PLoS Genet* 4, e1000110 (2008) and Srivastava et al., *Cell* 151, 1474-1487 (2012)]. Using this intrachromosomal

substrate, total NHEJ increased in cells lacking active BLM due to ML216 treatment (Figure 5B, S13A, text page 10, top). The experimental conditions for NHEJ assays using intrachromosomal substrates is mentioned in page 17, 3rd paragraph.

Comment #13:

Some data would be also performed BS cells. Are recruitment of these HR and NHEJ factors affected in BS cells?

Answer:

It is an interesting question about how the DDR pathway is mechanistically regulated in absence of BLM (for example in BS cells). We are actively pursuing this line of thought in multiple experimental conditions. I believe that will form part of a separate line of investigation and is beyond the scope of this manuscript.

Comment #14:

Moreover, the authors show that after 4-OHT, absence of BLM increased rates of sequences alterations, “providing BLM helicase functions affects c-NHEJ in G1 phase’. Overall, these data seem contradictory and interpretations quite confusing.

Answer:

We apologize in the earlier version of the paper we did not make the conclusions of the manuscript clearer. Figure 4G-4I demonstrates the effect of BLM helicase activity on the recruitment of the HR and NHEJ factors. On the other hand, Figure 5A-5E tells about the effect of BLM helicase activity on the repair processes, HR and NHEJ. To mechanistically distinguish between the two processes – we have differentiated between the two experimental conditions. In recruitment phase (for experiments done in Figure 4) the U2OS ASiSI-ER cells are continuously exposed to 4-OHT for 4 hrs. However in the repair phase (for experiments done in Figure 5A-5E), U2OS AsiSI-ER cells after first exposed to 4-OHT for 4 hrs, after which the 4-OHT is washed off and the repair is allowed to proceed (in presence of ML216, B02, SCR7) for 1 hr. We have specifically explained the change in the experimental conditions both in the text (page 9, bottom paragraph and page 15 bottom paragraph) and also in Figure S1.

In the recruitment phase BLM acts as a pro-repair protein as the absence of its helicase activity prevents the recruitment of the repair factors (Figure 4G-4J). However in the repair phase BLM acts as an anti-repair protein for both HR and NHEJ and thereby negatively regulates both HR and c-NHEJ (Figure 5A-5E). Hence we believe that we have been able to experimentally dissect and demonstrate the two functions of BLM during DDR. It is to be noted that based on in vitro results both the

pro-and anti-repair functions of BLM had been postulated earlier [Bugreev et al., *Genes and Dev*, 21, 3085-3094 (2007)]. This is the first in vivo validation of these two BLM functions with respect to DSBs. Based on the above reasoning we do not believe that the two datasets (Figure 4 and Figure 5A-5E) are contradictory. In the text (both results and discussion) we have simplified the language and have tried to explain our results in clear terms.

Comment #15:

However, different studies (in vitro using BS cells extracts, or by ligation of episomal plasmids upon BLM depletion, and even more recently by using intrachromosomal substrates) showing that BLM deficiency leads to a decrease of fidelity of end-joining have been already published (Gaymes, *Oncogene*, 2002 ; Grabarz, *Cell Report*, 2013...). Thus, results described in Table S3 are not really original. Anyway, number of sequenced clones in this study is not sufficient (15 for each conditions, comparison between 0, 1 or 2 clones on 15 total clones for insertions for instance is not really relevant !).

Answer:

Both papers referred by the reviewer (Gaymes et al., *Oncogene*, 2002; Grabarz et al., *Cell Report*, 2013) have used either plasmid or intrachromosomal substrates using extracts from BS cells or cells in which BLM has been depleted by siRNA. In contrast, this report will be the first time where alteration in the DSBs is being checked at annotated sites in the human genome without any exogenous transfection of substrates (Figure 5E of the revised manuscript).

Secondly BLM has many functions – many of which depend on the N- and C-terminal regions flanking the helicase domain of the protein. By using ML216 in these assays, we show for the first time that it is the helicase activity of BLM which allows it to negatively regulate c-NHEJ and thereby increase the rate of DNA alterations at specific DSBs in the human genome. Together we feel that our version of the assay is much nearer to the physiological conditions and aptly mimics the in vivo pathological situation as many of the BLM mutations in BS patients is in the helicase domain.

Finally we have taken into consideration the reviewer's concern about the number of clones which had been sequenced in the first version of the manuscript. We have now sequenced twenty-five individual clones for each for four DSBs (i.e. a total of 100 clones, data presented in Table S3). The combined data for DNA alterations for these 100 clones is shown in Figure 5E of the revised version.

Comment #15:

Page 11 “Co-depletion experiments show that the binding of BLM to the single-stranded DNA causes by...” should be replaced by “Co-depletion experiments suggest...” Here there is no data on direct binding of BLM on ssDNA, it is only suggested by the requirement of Mirin for its recruitment, thus potentially also the requirement of resection. Its binding could be through another factor on ssDNA.

Answer:

We agree with the reviewer’s viewpoint. We have made changes where we say that inhibition of BLM recruitment to DSBs by Mirin suggests the binding of BLM to single-stranded DNA. These changes are in Page 7, 2nd paragraph and Page 32, legend of Figure 5F.

Comment #16:

“This mechanism of BLM recruitment to DSBs is in contrast to the process characterized for its localization to stalled forks”. It would be interesting to summarize what is clearly different.

Answer:

We appreciate the reviewer’s question. We have elaborated in two different sections of the manuscript what is known about the mechanism of BLM recruitment during the generation of stalled forks – page 13 in Discussion, end of first paragraph and page 3 in Introduction, 2nd paragraph.

Comment #17:

Anti-recombination function of BLM is mediated by poly-ubiquitination (Böhm, 2014) has been reported. Here the authors show that poly-ubiquitylation of BLM is required for binding of NBS1. The data should be discussed in relation to what is known in the literature.

Answer:

We agree with the reviewer’s viewpoint. In the introduction we have incorporated the reference [Bohm and Bernstein, DNA Repair 22: 123–132 (2014)] referred to by the reviewer. In the Discussion we have (a) discussed about other ubiquitylation dependent mechanisms which affect the cellular functions of BLM (page 13, 1st

paragraph); (b) discussed about how other BLM post-translational modifications dependent mechanisms may work in coordination with the mechanisms of DNA repair which BLM regulates in different phases of the cell cycle (page 14, 2nd paragraph).

Comment #18:

“Expectedly BLM inhibits only c-NHEJ in G1 phase...” It has been reported that BLM counteracts MMEJ. This apparent discrepancy should be discussed.

Answer:

We thank the reviewer for pointing out the factual mistake. To avoid confusion we have removed the sentence from the revised version of the manuscript.

Reviewers' comments:

Reviewer #1 (Remarks to the Author):

This manuscript is now acceptable for publication. The clarification of the requirements for the two stages of recruitment and retention significantly enhances the significance of the study. Minor points raised previously have been dealt with adequately.

Reviewer #2 (Remarks to the Author):

BLM helicase plays an important role in recognition and resolution of DNA damage in replicative stress conditions. Mutation in BLM leads to cancer predisposition. Its recruitment during replication stress, an important source of genome instability, has been deeply studied. It is also important to well characterize its recruitment to double-strand breaks, to also elucidate its pro- or anti-repair function.

In this paper, Tripathi et al used the Diva system (an inducible site-specific double-strand breaks generating system) to analysed the recruitment of BLM on DSB. They show that (1) BLM is recruited from 80 bps to 9kb from the break site ; (2) BLM recruitment is independent of its own helicase activity but it is dependent of NBS1, MRE11 and ATM (as it has been previously reported) ; (3) BLM recruitment is abrogated upon MRE11 exonuclease activity inhibition and inhibition of RNF8 (which mediates the ubiquitination of BLM protein), (4) BLM is co-recruited with RAD51 in S-phase (as it has been reported upon replicative stress), and with XRCC4 in G1 phase, its helicase activity is important for the recruitment of key HR and c-NHEJ factors onto chomatin. The authors propose that BLM inhibits HR in S-phase and c-NHEJ in G1 phase. The authors propose also that ubiquitinated BLM interacts with NBS1.

The major novelties of this paper are : - the negative regulation of NHEJ by BLM in G1 phase and the proposed role of ubiquitination of BLM (by RNF8), indeed this BLM modification seems to be required for its interaction with NBS1.

The authors responded to most of the referee issues. The introduction and discussion have been improved.

However, there are still some concerns, especially about experiments done to analyse the DSBR regulation by BLM.

The authors did not use intrachromosomal substrates to analyse the efficiency of Homologous Recombination.

NHEJ has been measured by using transfected substrates at the same time than I-Sce1 plasmid. Cell lines stably established with intrachromosomal substrates have not been used. The level of I-Sce1 expression has not been verified.

The number of sequences analysed (by using DIVA system) is still not sufficient to calculate frequencies of deletions or insertions (for instance comparison of 1 to 3 insertions is not relevant).

The authors propose that NHEJ is increased upon BLM helicase activity inhibition (ML216). This increase seems depend entirely on the c-NHEJ, since ligase 4 inhibition (SCR7) abolished the NHEJ increase (Fig 5 and Fig S13) (by using an "extracts" assay). At the same time, BLM inhibition leads to a defect XRCC4 binding to DSBs in second wave (4 hours after 4OHT), considered to be the "DNA repair phase". The authors should discuss this point or repeat this experiment with intrachromosomal substrates (stably established).

The authors should discuss about the difference obtained with the ATM depletion (siRNA) or with the ATM inhibitor (KU5933). Depletion of ATM leads to a defect of BLM (30 min and 4 hours after

induction of DNA breaks). In contrast ATM inhibitor leads only to a defect of BLM recruitment to DSB in the first phase (30 min after 4-OHT).

Answers to reviewer's comments

Reviewer #2:

Comment #1:

However, there are still some concerns, especially about experiments done to analyse the DSB regulation by BLM. The authors did not use intrachromosomal substrates to analyse the efficiency of Homologous Recombination.

Answer:

We agree in principle with the reviewer's concern about not using intrachromosomal substrates to analyse the efficiency of Homologous Recombination. However, the cells with integrated HR substrates were not readily made available to us. The role of BLM activity in controlling HR is very well known in literature. For our data in Figure 5A we have used in vitro substrates for HR which have been previously used and accurately recapitulates the negative regulatory role of BLM activity on HR during S-phase.

Comment #2:

NHEJ has been measured by using transfected substrates at the same time than I-Sce1 plasmid. Cell lines stably established with intrachromosomal substrates have not been used. The level of I-Sce1 expression has not been verified. The number of sequences analysed (by using DIVA system) is still not sufficient to calculate frequencies of deletions or insertions (for instance comparison of 1 to 3 insertions is not relevant).

Answer:

Again we agree in principle with the reviewer's concern about not using intrachromosomal substrates to analyse the efficiency of NHEJ. However, the cells with integrated NHEJ substrates were not readily available to us. Hence we have used three independent lines of evidence to demonstrate that BLM indeed inhibits c-NHEJ in a cell cycle dependent manner. These include (a) a transfection based NHEJ assay which uses intrachromosomal substrates (Figure 5B, Supplementary Figure S13A-S13C); (b) an in vitro assay using c-NHEJ substrates and detecting joined product formation (Figure 5C, Supplementary Figure S13D, S13E) and (c) a Sanger sequencing based assay which detected the changes in the nucleotide incorporated in the genomic DNA around four annotated DSB sites (Figure 5E, Supplementary Table S3). Further, to address reviewer's concern, we have presented additional data (Supplementary Figure S13B and S13C of the present version) indicating that the

transfection efficiency was equal in all conditions in which the intrachromosomal assays for c-NHEJ were carried out (Figure 5B).

To strengthen the data regarding identification of sequence alteration in absence of BLM helicase activity, additional sequencing of the DSBs in both –ML216 and +ML216 was carried out (Figure 5E, Supplementary Table S3). In the present version data for thirty-five individual clones for each of the four DSBs in both –ML216 and +ML216 experimental conditions have been represented i.e. a total of two-hundred and eighty sequencing of the DSBs were done. It is to be noted that we have progressively increased the number of clones being sequenced for each of the four DSBs. We agree with the reviewer's point that for certain DSBs for a particular type alteration the number of nucleotide changes is sometimes in single digits (as seen in Supplementary Table S3). However, we believe that it is essential to analyze the data in a cumulative manner for all the four tested DSBs to understand whether the percentage of the insertions, deletions and deletions with insertions are altered in absence of BLM helicase activity. This data is represented in Figure 5E which demonstrates that absence of BLM helicase activity statistically increased the rates of sequence alterations (involving deletions, insertions and deletions with insertions) at or near the AsiSI junctions.

Comment #3:

The authors propose that NHEJ is increased upon BLM helicase activity inhibition (ML216). This increase seems depend entirely on the c-NHEJ, since ligase 4 inhibition (SCR7) abolished the NHEJ increase (Fig 5 and Fig S13) (by using an "extracts" assay). At the same time, BLM inhibition leads to a defect XRCC4 binding to DSBs in second wave (4 hours after 4OHT), considered to be the "DNA repair phase". The authors should discuss this point or repeat this experiment with intrachromosomal substrates (stably established).

Answer:

To further clarify the point that BLM has dual functions as both a pro-recombinogenic and anti-recombinogenic protein we have added the following paragraph in the discussion section (page 14):

“To mechanistically understand how BLM has both a pro-recombinogenic role and also an anti-recombinogenic function, cells were grown in presence of 4-OHT for 4 hrs and then for 1 hr in media without any 4-OHT (schematic in Supplementary Figure S1). This 1 hr time period was vital as during this phase the recruitment phase will ceased exist, the repair phase was initiated at the DSBs and consequently the anti-recombinogenic function of BLM became predominant. Hence while BLM activity was essential for the recruitment of HR and c-NHEJ factors in a cell cycle specific manner during the recruitment phase (Figure 4G-4J, Supplementary Figure S12), the

same BLM activity inhibited HR and c-NHEJ (Figure 5A-5E) in the repair phase due to the switch of BLM from a pro- to anti-recombinogenic role”.

The difference in the experimental protocol between the recruitment phase and the repair phase of BLM has also been detailed in the result section (page 9-10) and also in the methods section (page 16).

Comment #4:

The authors should discuss about the difference obtained with the ATM depletion (siRNA) or with the ATM inhibitor (KU55933). Depletion of ATM leads to a defect of BLM (30 min and 4 hours after induction of DNA breaks). In contrast ATM inhibitor leads only to a defect of BLM recruitment to DSB in the first phase (30 min after 4-OHT).

Answer:

We appreciate the point made by the reviewer. We have discussed this in the discussion (page 13) of the revised manuscript where we have stated:

“It is to be noted that while depletion of ATM leads to abrogation of BLM recruitment after both 30 min and 4 hrs of 4-OHT treatment, inhibition of ATM activity has its effect only at the earlier time point (Figure 2C, 2E, Supplementary Figure S7, S8D). This indicates that at the later time periods of DSB induction, BLM which is already at the DSBs do not directly need the kinase activity of ATM. Instead the kinase activity of ATM in this time period is probably needed by its other substrates of the DDR pathway (like MRN complex). However, the physical presence of ATM is necessary as without it these other substrates will not get phosphorylated and the DDR pathway cease to be effective”.

REVIEWERS' COMMENTS:

Reviewer #2 (Remarks to the Author):

BLM helicase plays an important role in recognition and resolution of DNA damage in replicative stress conditions. Mutation in BLM leads to cancer predisposition. Its recruitment during replication stress, an important source of genome instability, has been deeply studied. It is also important to well characterize its recruitment to double-strand breaks, to also elucidate its pro- or anti-repair function.

In this paper, Tripathi et al used the Diva system (an inducible site-specific double-strand breaks generating system) to analysed the recruitment of BLM on DSB. They show that (1) BLM is recruited from 80 bps to 9kb from the break site ; (2) BLM recruitment is independent of its own helicase activity but it is dependent of NBS1, MRE11 and ATM (as it has been previously reported) ; (3) BLM recruitment is abrogated upon MRE11 exonuclease activity inhibition and inhibition of RNF8 (which mediates the ubiquitination of BLM protein), (4) BLM is co-recruited with RAD51 in S-phase (as it has been reported upon replicative stress), and with XRCC4 in G1 phase, its helicase activity is important for the recruitment of key HR and c-NHEJ factors onto chomatin. The authors propose that BLM inhibits HR in S-phase and c-NHEJ in G1 phase. The authors propose also that ubiquitinated BLM interacts with NBS1.

The major novelties of this paper are : - the negative regulation of NHEJ by BLM in G1 phase and the proposed role of ubiquitination of BLM (by RNF8), indeed this BLM modification seems to be required for its interaction with NBS1.

The authors responded to most of the referree issues.

Although, cell lines stably established with intrachromosomal substrates have not been used, the authors added (figure S13), additional crucial control such as the efficiency of tranfection and especially they show a comparable I-SceI protein level with or without ML216 treatment. They performed also additional sequencing of the DSBs in both ML216 and +ML216 was carried out reinforcing their data (Table S3). Supplementary figures with sequences could be shown.

The introduction and discussion have been improved. Especially, they added paragraphes (page 13 and page 14) to discuss the comment 2 and 3 of the reviewer.

Concerning, the comment 1, it is still a pity that authors did not use intrachromosomal substrates to analyse the efficiency of Homologous Recombination.

Answers to reviewer's comments

Reviewer #2:

Comment:

Although, cell lines stably established with intrachromosomal substrates have not been used, the authors added (figure S13), additional crucial control such as the efficiency of transfection and especially they show a comparable I-SceI protein level with or without ML216 treatment. They performed also additional sequencing of the DSBs in both ML216 and +ML216 was carried out reinforcing their data (Table S3). Supplementary figures with sequences could be shown.

The introduction and discussion have been improved. Especially, they added paragraphs (page 13 and page 14) to discuss the comment 2 and 3 of the reviewer.

Concerning, the comment 1, it is still a pity that authors did not use intrachromosomal substrates to analyse the efficiency of Homologous Recombination.

Answer:

We thank the reviewer for approving the changes and the additional data incorporated in the previous version of the manuscript.

In response to reviewer's request we have now added Supplementary Notes where we have given the raw sequences from which the data was extracted to obtain Figure 5E and Supplementary Table S3. Please do note that we have incorporated into Supplementary Notes only those sequences in which alterations (involving deletions, insertions and deletions with insertions) at or near the AsiSI junctions were found.

We agree in principle with the reviewer's concern about not using intrachromosomal substrates to analyze the efficiency of Homologous Recombination and Non Homologous End Joining. However, the cells with integrated HR and NHEJ substrates were not readily made available to us.